# Claudin5 protects the peripheral endothelial barrier in an organ and vessel-type-specific manner

Mark Richards[1,2]*, Emmanuel Nwadozi[1,2], Sagnik Pal[1,2], Pernilla Martinsson[1,2], Mika Kaakinen[3], Marleen Gloger[1,2], Elin Sjöberg[1,2], Katarzyna Koltowska[1,2], Christer Betsholtz[1,4], Lauri Eklund[3], Sofia Nordling[1,2]†, Lena Claesson-Welsh[1,2]*†

[1]Department of Immunology, Genetics and Pathology, Uppsala University, Uppsala, Sweden; [2]Beijer Gene- and Neuro Laboratory and Science for Life Laboratories, Uppsala University, Uppsala, Sweden; [3]Oulu Center for Cell-Matrix Research, Faculty of Biochemistry and Molecular Medicine, Biocenter Oulu, University of Oulu, Oulu, Finland; [4]Department of Medicine Huddinge, Karolinska Institutet, Campus Flemingsberg, Neo, Huddinge, Sweden

**\*For correspondence:**
mark.richards@igp.uu.se (MR);
lena.welsh@igp.uu.se (LC-W)

†These authors contributed equally to this work

**Competing interest:** The authors declare that no competing interests exist.

**Abstract** Dysfunctional and leaky blood vessels resulting from disruption of the endothelial cell (EC) barrier accompanies numerous diseases. The EC barrier is established through endothelial cell tight and adherens junctions. However, the expression pattern and precise contribution of different junctional proteins to the EC barrier is poorly understood. Here, we focus on organs with continuous endothelium to identify structural and functional in vivo characteristics of the EC barrier. Assembly of multiple single-cell RNAseq datasets into a single integrated database revealed the variability and commonalities of EC barrier patterning. Across tissues, Claudin5 exhibited diminishing expression along the arteriovenous axis, correlating with EC barrier integrity. Functional analysis identified tissue-specific differences in leakage properties and response to the leakage agonist histamine. Loss of Claudin5 enhanced histamine-induced leakage in an organotypic and vessel type-specific manner in an inducible, EC-specific, knock-out mouse. Mechanistically, Claudin5 loss left junction ultrastructure unaffected but altered its composition, with concomitant loss of zonula occludens-1 and upregulation of VE-Cadherin expression. These findings uncover the organ-specific organisation of the EC barrier and distinct importance of Claudin5 in different vascular beds, providing insights to modify EC barrier stability in a targeted, organ-specific manner.

## Editor's evaluation

Understanding molecular and cellular mechanisms for endothelial cell (EC) barrier integrity at homeostasis and its impairment at pathologic conditions are fundamental and central topics for understanding vascular biology and related cardiovascular diseases. In this study, the authors elegantly demonstrated that the claudin5 deficiency enhances histamine-induced leakage in organ- and vessel type-specific, and size-selective manners, which could be the result of alternative compositions of adherens and tight junctional proteins in the ECs. This study will aid our ability to modify EC barrier stability in a targeted, organ-specific manner.

## Introduction

Blood vessel dysfunction is a hallmark and contributing factor to the progression of numerous pathologies including solid tumours, ischemic diseases and inflammatory conditions (*Lee et al., 2007*; *Burke*

*and Miles, 1958*; *Hashizume et al., 2000*). In such pathologies, local concentrations of inflammatory cytokines and growth factors are increased, leading to weakening of the endothelial cell (EC) barrier, a regulatable interface between circulating blood and the surrounding tissue environment (*Senger et al., 1983*; *Miles and Miles, 1952*; *Palade et al., 1979*). Loss of EC barrier integrity in-turn leads to enhanced molecular and cellular passage across the endothelium resulting in edema, tissue damage, atrophy, and disease progression (*Wu et al., 2014*; *Fleckenstein et al., 2018*). EC barrier integrity is mediated primarily by adherens junctions (AJs) and tight junctions (TJs), composed of transmembrane proteins forming intercellular interactions that bridge adjacent EC membranes (*Corada et al., 1999*; *Claesson-Welsh et al., 2021*). These transmembrane proteins also associate with intracellular scaffolding and signaling proteins, as well as the cytoskeleton, which regulate their localisation and stability. The composition and organisation of AJs and TJs thus determines the relative strictness and regulatable nature of the EC barrier.

AJs are ubiquitously distributed throughout the vascular system in all vessel subtypes (arterial, capillary, and venous) and consist of the largely endothelial-specific transmembrane protein vascular endothelial (VE)-Cadherin, which forms homophilic interactions between cells and associates intracellularly with the actin cytoskeleton via several members of the catenin family (*Bazzoni and Dejana, 2004*). VE-cadherin-catenin interactions are highly regulated by their phosphorylation status, which modulates AJ stability and integrity of junctions (*Smith et al., 2020*; *Eliceiri et al., 1999*; *Orsenigo et al., 2012*).

TJs are also ubiquitously distributed but are in comparison relatively less well defined, and their composition and localisation appear more heterogeneous in comparison to AJs. Numerous transmembrane proteins are associated with TJs, including members of the claudin family, the tight junction-associated MARVEL proteins, such as Occludin, the junction-associated molecule (JAM) family, and endothelial-cell selective adhesion molecule (ESAM) (*Greene et al., 2019*; *Nasdala et al., 2002*; *Martin-Padura et al., 1998*; *Raleigh et al., 2010*). With few exceptions, TJ proteins are broadly expressed and not unique to vascular ECs. Similar to AJs, TJ proteins exist in homophilic complexes between adjacent ECs, as well as with intracellular scaffolding proteins including members of the zonula occludens (ZO) family and Cingulin (*Stevenson et al., 1986*; *Schossleitner et al., 2016*). The relative proportion of these proteins within junctions is believed to influence barrier strictness, however little is known about the actual composition of TJs and how each component contributes to EC barrier integrity, particularly in vivo.

Of the TJ proteins, Claudin5 is the best studied regarding its contribution to EC barrier integrity. Claudin5 is highly expressed in the brain vasculature and is an important component of the blood brain barrier (BBB) and blood retinal barrier (BRB) where it is responsible for restricting the passage of small molecules (*Daneman et al., 2010*). Consequently, constitutive *Cldn5* knock-out in mice results in aberrant BBB permeability and death 10 hours after birth (*Nitta et al., 2003*). Furthermore, down-regulation of Claudin5 expression, and subsequent enhanced EC permeability, is associated with neurological disorders such as multiple sclerosis, stroke, epilepsy and schizophrenia (*Alvarez et al., 2011*; *Knowland et al., 2014*; *Yan et al., 2018*; *Sun et al., 2004*). Outside of the CNS, Claudin5 is expressed in ECs to a lower degree, as are other TJ proteins such as Occludin (*Scalise et al., 2021*). Consequently, non-CNS blood vessels have a more permeable EC barrier, although EC permeability also differs greatly between these peripheral tissues and is inherently linked to organ-specific function (*Richards et al., 2021*; *Augustin and Koh, 2017*). Furthermore, EC barrier integrity differs between vessel subtypes within these vascular beds (*McDonald, 1994*; *Honkura et al., 2018*). Typically, the arteriolar aspect of the vasculature is more resistant to EC barrier disruption than the venular side. Interestingly, at least in the mouse ear dermis, EC barrier integrity correlates with Claudin5 expression, which is evident in arterioles but progressively diminishes in capillaries and venules (*Honkura et al., 2018*). The expression pattern of Claudin5 in different vessel subtypes is however poorly understood in tissues outside the CNS, as is the arrangement and differential impact of other TJ proteins on EC barrier integrity. An important goal in clinical medicine is to suppress disease progression by dampening vascular leakage; on the other hand, improved drug delivery across the EC barrier would improve the outcome in a range of diseases. Therefore, better understanding of EC barrier composition will enable the development of tools to both open and close the barrier at will.

In this study, analysis of single-cell RNAseq (scRNAseq) datasets from multiple peripheral vascular beds demonstrates that in general, EC junction proteins have a similar distribution in tissues with

a continuous endothelium, although with subtle variability in TJ patterning. In all tissues examined here; ear skin, back skin, trachea, skeletal muscle and heart, *Cldn5* exhibits decreasing expression along the arteriovenous axis. We show that *Cldn5* expression inversely correlates with histamine-induced vascular leakage, although precise patterning exhibits tissue-specific variability. Moreover, analysis of Claudin5 function in adult mice using an inducible, endothelial-specific, knock-out mouse shows organotypic differences, with weakening of the EC barrier occurring to different degrees. Ultra-structurally, loss of Claudin5 has no effect on EC junction organisation but leads to changes in the expression of other junction proteins including VE-Cadherin, ZO-1, and Occludin. Together, these data uncover the organotypic heterogeneity that exists in the organisation of EC junctions and the importance of TJ components for general EC barrier integrity.

## Results

### Patterning of the EC barrier at the single-cell level

The organotypic properties of EC junctions were first addressed by exploring the expression profiles of junctional genes in publicly available murine scRNAseq datasets of heart, skeletal muscle and tracheal blood vascular ECs (BECs) (*Kalucka et al., 2020*; *Tabula Muris Consortium, 2020*) in combination with newly generated scRNAseq data of mouse dermal BECs (*Figure 1—figure supplement 1*). In order to ensure maximum comparability between vessel subsets (i.e. arterial, capillary, and venous) in different organs, these datasets were integrated using mutual nearest neighbor (MNN) alignment to correct for differences between similar populations, followed by trajectory inference using the tSpace algorithm (*Figure 1A*; *Haghverdi et al., 2018*; *Dermadi et al., 2020*). An isolated trajectory spanning from arterial to venous BECs of the integrated data (*Figure 1B*) was subjected to equidistant binning (*Dermadi et al., 2020*) and the expression of vessel subset-specific markers were subsequently used to guide the annotation of the bins as arterial, arterial/capillary, capillary, capillary/venous, or venous (*Figure 1C and D*; *He et al., 2018*; *Vanlandewijck et al., 2018*; *Kalucka et al., 2020*; *Brulois et al., 2020*). Subset allocation and clustering was validated by stand-alone analysis of each individual organ and by comparison to previously published cluster annotations when available (*Figure 1—figure supplement 2*; *Kalucka et al., 2020*). This analysis provides a comprehensive integrated database of gene expression across comparable BEC subsets in multiple tissues.

Subsequently, the expression of genes associated with EC junctions was investigated in each organ and vessel subset (*Figure 1E*, *Figure 1—figure supplement 3*). A differential gene expression analysis between each subset and all other cells in an organ was utilized to guide conclusions (*Figure 1—source data 1–4*). The adherens junction gene *Cdh5* (VE-Cadherin) was ubiquitously expressed within each organ but was generally slightly lower in venous vessels compared to other subsets within these tissues (*Figure 1E*). Moreover, N-Cadherin (*Cdh2*) expression was relatively low, whereas T-Cadherin (*Cdh13*) was high, in all organs (*Figure 1—figure supplement 3*). The cell adhesion molecule Caecam1 was also relatively abundantly expressed in all organs. Meanwhile, the Nectins 1, 2 and 3 showed expression restricted to skeletal and heart muscle ECs. As expected, intracellular junction proteins such as α-, β- and p120-catenin (*Ctnna1*, *Ctnnb1*, *Ctnnd1*, respectively) were ubiquitously expressed, as were ZO-1 (*Tjp1*) and ZO-2 (*Tjp2*) and the angiomotin-like proteins 1 and 2 (*Amotl1* and *Amotl2*) (*Figure 1—figure supplement 3*).

Tight junction genes displayed distinct expression patterns. The JAM family of junctional adhesion molecules (*F11r, Jam2, Jam3*) exhibited reasonably homogenous expression, but was occasionally more abundantly expressed in capillaries than in arteries and veins, in an organotypic manner (*Figure 1E*, *Figure 1—figure supplement 3*). *Esam* similarly was homogenously expressed with reduced expression in the venous subset of some tissues (*Figure 1E*). Expression of the tight junction-associated Marvel proteins (TAMPs) *Ocln*, *Marveld2*, and *Marveld3* was relatively low, being limited to only a few BECs in an organotypic and subset-specific manner (*Figure 1E*, *Figure 1—figure supplement 3*). Most members of the Claudin family were expressed at a very low level with the exception of *Cldn5*, *Cldn15,* and *Cldnd1,* the latter of which was specifically absent in the tracheal vasculature. Both *Cldn5* and *Cldn15,* however, exhibited diminishing expression along the arteriovenous axis, with *Cldn5* in particular exhibiting higher expression in arteries compared to veins, displaying fold changes between 3 (ear skin) and 15 (skeletal muscle) (*Figure 1E and F*). Accordingly, analysis

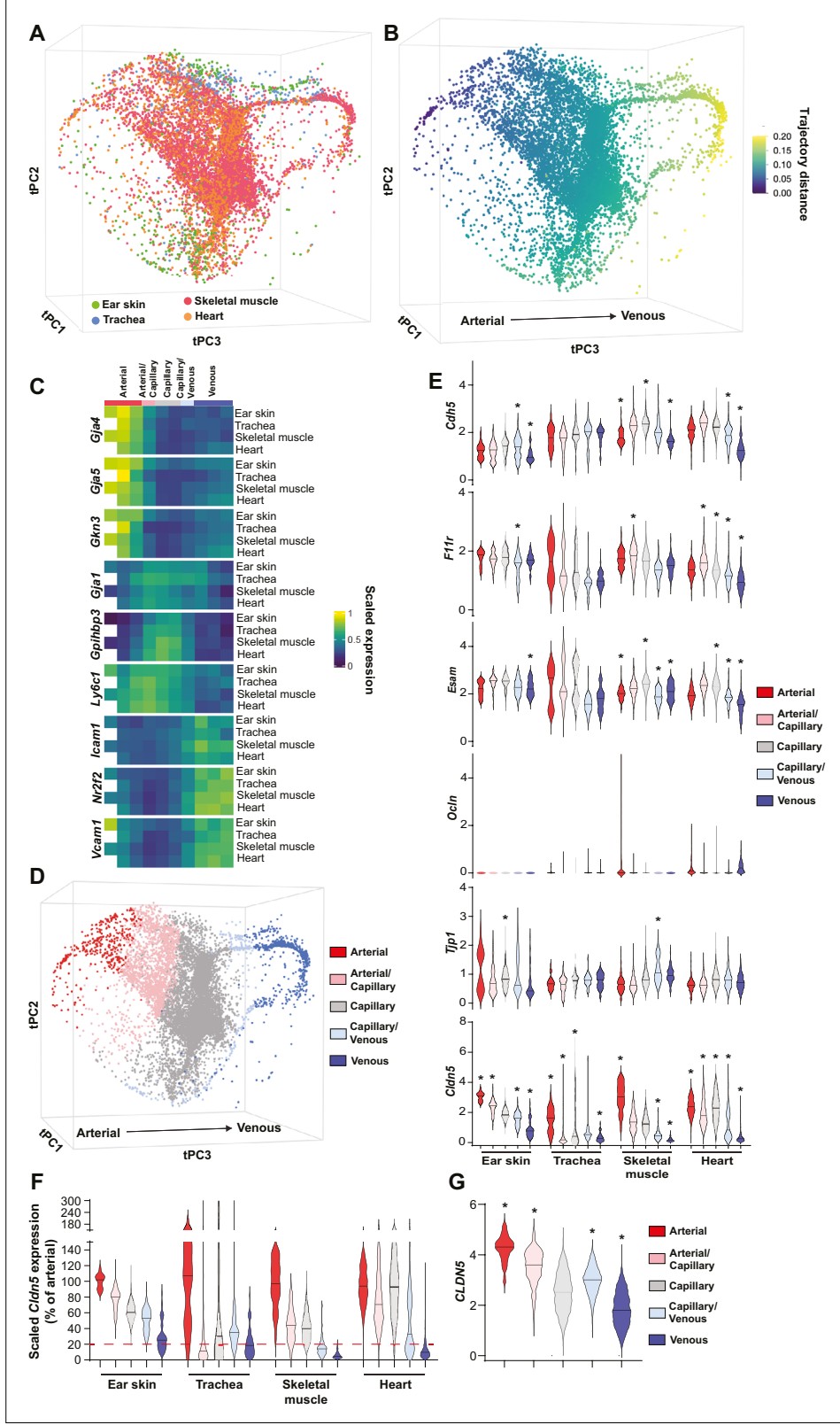

**Figure 1.** Patterning of the EC barrier at the single-cell level. (**A**) Principal component analysis of the distances within 400 trajectories calculated with integrated data of murine datasets of ear skin, trachea, skeletal muscle, and heart blood endothelial cells (BECs). Colors illustrate the distribution of BECs (CD31+/CD45-/Lyve1-) for each organ. (**B**) Principal component analysis of trajectory distances colored by the distance along an isolated trajectory

*Figure 1 continued on next page*

*Figure 1 continued*

spanning from arterial to venous BEC. (**C**) Mean gene expression for each organ after equidistant binning of the isolated trajectory shown in B. Supervised vessel subset specifications (Top) based on the expression of previously established marker genes. (**D**) Principal component analysis of trajectory distances colored by the vessel subsets defined in C. (**E**) Violin plots of gene expression for BEC junctional components. Gene expression was normalized to account for differences in sample library size and has been imputed to account for dropouts in the data as described in Materials and methods. (**F**) *Cldn5* expression in murine BEC datasets scaled per organ according to the mean expression in the arterial BECs of each organ. Red dashed line represents a fivefold reduction in expression compared to arterial BECs. (**G**) *CLDN5* expression in human dermal BECs. n=534 ear skin, 559 trachea, 3498 skeletal muscle, 6423 heart and 8518 human BEC. * denotes statistical significance following differential gene expression analysis (*Figure 1—source data 1–5*).

The online version of this article includes the following source data and figure supplement(s) for figure 1:

**Source data 1.** Spreadsheets detailing the results of the differential gene expression analysis conducted between mouse BEC subtypes in ear skin.

**Source data 2.** Spreadsheets detailing the results of the differential gene expression analysis conducted between mouse BEC subtypes in trachea.

**Source data 3.** Spreadsheets detailing the results of the differential gene expression analysis conducted between mouse BEC subtypes in skeletal muscle.

**Source data 4.** Spreadsheets detailing the results of the differential gene expression analysis conducted between mouse BEC subtypes in heart.

**Source data 5.** Spreadsheet detailing the results of the differential gene expression analysis conducted between human dermal BEC subtypes.

**Figure supplement 1.** Gating strategy for the FACS isolation of single blood vessel BECs from the mouse ear skin.

**Figure supplement 2.** BEC subset allocation and clustering of individual mouse organs.

**Figure supplement 3.** Expression of mouse endothelial junctional components.

**Figure supplement 4.** Analysis of human skin BECs.

**Figure supplement 5.** Expression of human dermal endothelial junctional components.

of genes differentially expressed between vessel subsets showed consistently significant changes in *Cldn5* expression (*Figure 1—source data 1–4*).

We further utilized a recently published human skin BEC scRNAseq dataset to investigate whether EC junctional patterning is faithfully conserved between human and mouse (*Li et al., 2021*). Similar analysis of human skin BECs as employed for mouse BECs (*Figure 1—figure supplement 4*) showed enrichment of adherens junction genes *CDH5* and *CDH13* in the arterial and capillary subsets, with lower levels in venous proximal capillaries and veins (*Figure 1—figure supplement 5*). In keeping with the overall trend in the mouse vasculature, human tight junction component expression was also relatively higher in arteries and/or capillaries; *F11R*, *ESAM*, *CLDN5*, and *CLDN10* expression was significantly higher in arteries and downregulated in veins (*Figure 1G*, *Figure 1—figure supplement 5*, *Figure 1—source data 5*).

## Organotypic regulation of barrier integrity

In the ear dermis, expression of Claudin5 inversely correlates with vessel susceptibility to Vascular Endothelial Growth Factor A (VEGF-A)-induced leakage (*Honkura et al., 2018*). We thus decided to examine whether EC barrier integrity was similarly patterned in other tissues and in response to other agonist classes such as inflammatory cytokines. For this purpose, we employed histamine in an intravital imaging setup coupled with atraumatic intradermal agonist injection, in which acute leakage may be accurately assessed (*Honkura et al., 2018*). Claudin5 expression was ascertained through *Cldn5* promoter-driven expression of GFP (*Cldn5*(BAC)-GFP) (*Honkura et al., 2018*). Here, intradermal administration of histamine elicited a similar leakage patterning as previously shown for VEGF-A, with leakage occurring only in vessels possessing ECs lacking apparent *Cldn5*(BAC)-GFP expression (*Figure 2A and B* and *Video 1*). Interestingly, within the leakage-susceptible vasculature, a population of *Cldn5*(BAC)-GFP-negative vessels failed to respond to histamine stimulation, highlighting the potential importance of factors other than Claudin5 that influence vessel susceptibility to stimulation (*Figure 2C*; *Richards et al., 2021*).

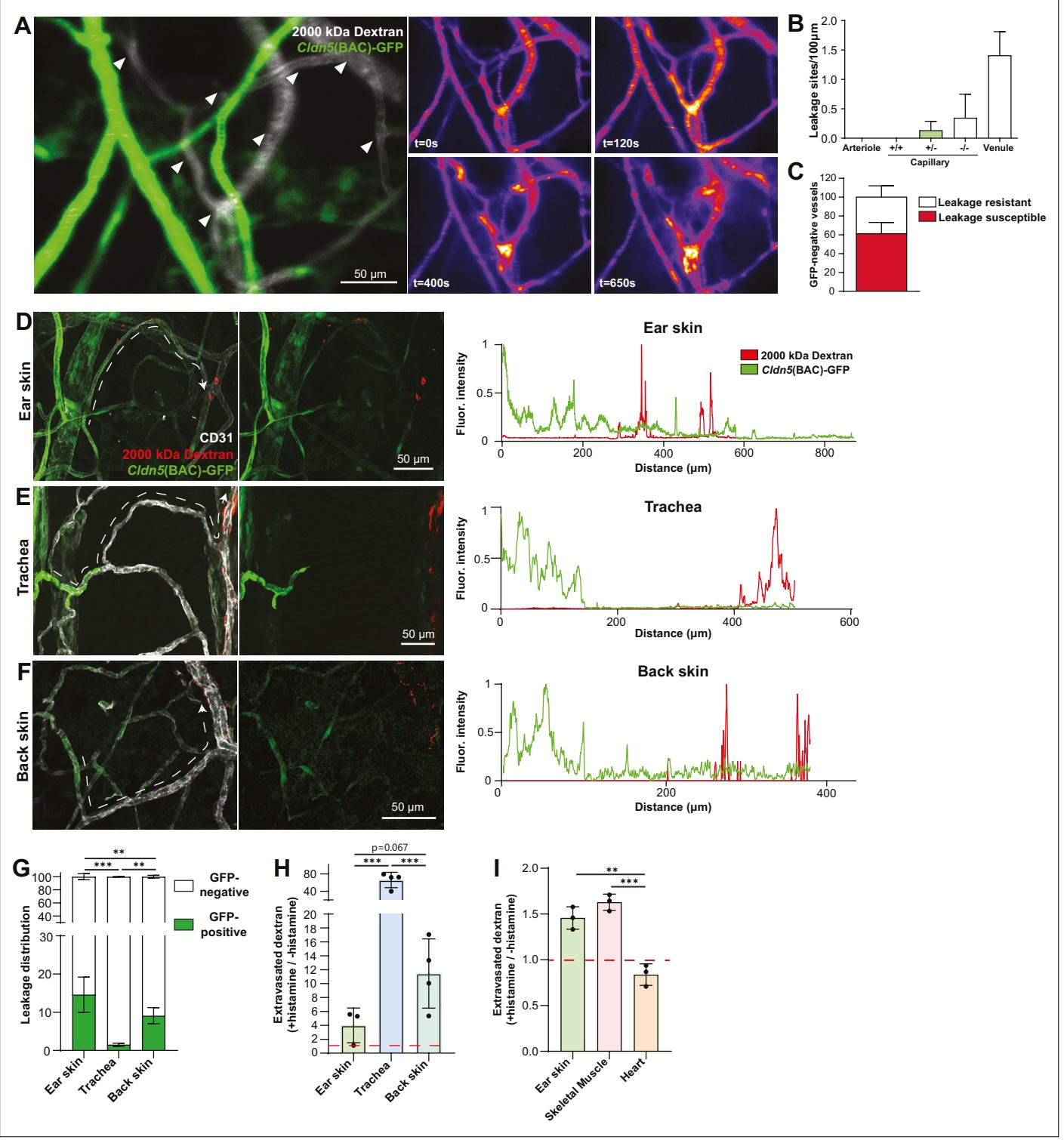

**Figure 2.** Organotypic integrity of the EC barrier. (**A**) Leakage patterning in *Cldn5*(BAC)-GFP mouse ear skin in response to intradermal histamine. Left, overlay of *Cldn5*(BAC)-GFP-positive and -negative vessels (visualised through circulating TRITC dextran). Arrowheads show sites of leakage. Right, stills of leakage in the vasculature shown on the left following intradermal histamine stimulation. (**B**) Leakage sites per vessel length in different vessel categories. +/+denotes capillary segments with full GFP expression,+/-denotes capillary segments with mixed GFP expression, -/- denotes capillary segments with no GFP expression. n=4, 2 or more acquisitions/mouse. (**C**) Proportion of *Cldn5*(BAC)-GFP-negative vessels susceptible or resistant to leakage. n=4, 2 or more acquisitions/mouse. (**D–F**) Leakage patterning in the ear skin (**D**), trachea (**E**) and back skin (**F**) in response to the systemic administration of histamine. Left, representative image. Dashed line shows progression of a blood vessel from arteriolar to venular. Right, representative

*Figure 2 continued on next page*

*Figure 2 continued*

fluorescent intensity line profile of *Cldn5*(BAC)-GFP and TRITC 2000 kDa dextran along the dashed line (Left). **(G)** Proportion of 2000 kDa FITC leakage area that occurs in vessels that are *Cldn5*(BAC)-GFP-positive (contain some positive cells) and *Cldn5*(BAC)-GFP-negative (contain no positive cells) in ear skin, back skin and trachea. n≥3 mice, 3 or more fields of view/mouse. **(H)** Fold change in 2000 kDa TRITC dextran extravasation from leakage permissive vessels in ear skin, back skin and trachea with and without systemic histamine stimulation. Dashed line represents unstimulated tissue. n=3 mice, 3 or more fields of view/mouse. **(I)** Fold change in tissue 2000 kDa FITC dextran following systemic histamine stimulation and formamide extraction of ear skin, skeletal muscle and heart. Dashed line represents unstimulated tissue. n=3 mice. Error bars; mean ± SD. Statistical significance: one-way ANOVA with Tukey's post-hoc test (multiple comparisons; **G–I**).

The online version of this article includes the following figure supplement(s) for figure 2:

**Figure supplement 1.** Histamine leakage in skeletal muscle and heart vasculature.

Next, a model of acute systemic leakage was used whereby histamine, when injected into the circulation via the tail-vein, results in widespread disruption of EC junctions and leakage of fluorescent tracers into the parenchyma (*Richards et al., 2021*). When given systemically, along with a 2000 kDa lysine-fixable dextran, histamine caused leakage in the ear dermis from venules and also from capillaries exhibiting a mixed expression of *Cldn5*(BAC)-GFP (*Figure 2D*, left). Skeletal muscle also showed leakage susceptibility starting in the capillary bed where *Cldn5*(BAC)-GFP expression was heterogeneous (*Figure 2—figure supplement 1*, left). The heart vasculature meanwhile showed no extravascular dextran accumulation with or without histamine stimulation (*Figure 2—figure supplement 1*, right). Unlike the ear dermis and skeletal muscle, the tracheal vasculature displayed a clear separation between *Cldn5*(BAC)-GFP-positive arterioles and leakage susceptible vessels, with the capillaries that cross the cartilage rings being *Cldn5*(BAC)-GFP-negative and resistant to stimulation (*Figure 2E*, left). Interestingly, examination of the mouse back skin revealed an intermediary phenotype, with vessels possessing a mixed expression of *Cldn5*(BAC)-GFP being largely resistant to leakage, whilst immediately subsequent venules exhibited a strong leakage phenotype (*Figure 2F*, left). Representative line profiles demonstrate this variable patterning, with extravasated dextran interspersed with a *Cldn5*(BAC)-GFP-positive signal in the ear dermis (*Figure 2D*, right) but not in the back skin or trachea (*Figure 2E and F*; right). These differing leakage patterns were also apparent from the increased proportion of extravasated dextran from *Cldn5*(BAC)-GFP-positive versus *Cldn5*(BAC)-GFP-negative vessels in the ear skin compared to back skin and trachea (*Figure 2G*). Additionally, these tissues differed in their magnitude of leakage to histamine stimulation (*Figure 2H*). In response to the same stimulus, leakage permissive vessels in the trachea and the back skin showed more extensive leakage as compared to the ear skin. Skeletal muscle showed a similar leakage response as the ear skin whilst, as expected, no histamine-induced leakage was observed in the heart (*Figure 2I*).

This data shows that outside of the CNS, a zonation of Claudin5 expression exists along the arteriovenous axis. Moreover, the relationship between Claudin5 expression and susceptibility to vascular leakage differs between vascular beds, particularly in capillaries, as does the sensitivity of ECs to stimulation and their resulting junctional disruption.

## Claudin5 exhibits organotypic protection of the EC barrier

Claudin5 is a well-known determinant of blood vessel barrier integrity in the CNS. Constitutive Claudin5 deficiency results in the passage of small molecules from the blood into the cerebrospinal fluid and death shortly after birth (*Nitta et al., 2003*). Nothing is known about the role that Claudin5 plays in barrier stability outside of the CNS. To investigate this, we used an inducible, EC-specific *Cldn5* loss-of-function mouse model (*Cldn5*^fl/fl;*Cdh5*^CreERT2, *Cldn5* iECKO)

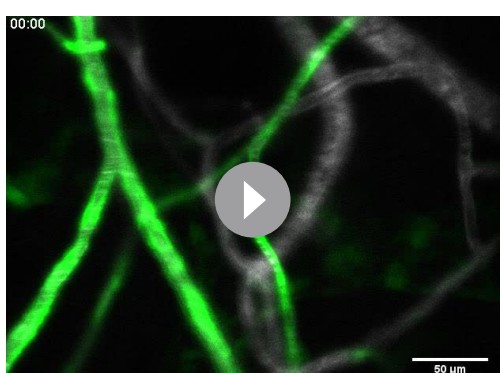

**Video 1.** Histamine-mediated leakage in *Cldn5*(BAC)-GFP mice. Extravasation of circulating 2000 kDa TRITC Dextran (pseudocolor) after intradermal injection of histamine in the ear dermis of *Cldn5*(BAC)-GFP mice. The first 30 frames show a still at t=0 to show *Cldn5*(BAC)-GFP expression (green) overlaid with 2000 kDa Dextran (grey).

https://elifesciences.org/articles/78517/figures#video1

(*Figure 1—figure supplement 1A*). Systemic administration of tamoxifen to these 8–16 weeks old mice (*Figure 3A*) led to no obvious physiological or behavioral defects the first five days following treatment completion. Analysis of lung lysates showed a 60% reduction in Claudin5 protein levels and a near total loss of *Cldn5* RNA in iECKO mice compared to controls (*Figure 3B-D*). Blood vessels in the ear dermis meanwhile showed an approximately 75% reduction in Claudin5 levels, as discerned by immunofluorescent staining, with some remaining protein expression evident in arterioles (*Figure 3E*).

*Cldn5* iECKO mice were next investigated to assess whether Claudin5 maintains the permeability of blood vessels in tissues outside of the CNS. To study this, mice were systemically injected with differently sized fluorescent dextrans, which were allowed to circulate before mice were perfused and organs collected. Extravasated dextran was subsequently extracted into formamide and measured according to their spectra. Initially, to better understand tissue-specific differences in the EC barrier, organs were assessed for their basal permeabilities to 10 and 70 kDa dextran (*Figure 3F*). Interestingly, the back skin showed relatively high basal permeability when compared with ear skin, which showed minor dextran extravasation, while there was no basal leakage from skeletal muscle and heart. This basal permeability was unaffected by loss of Claudin5 when comparing *Cldn5* iECKO with Cre-negative control mice, for all sizes of dextran investigated (*Figure 3G*).

We next sought to determine whether the loss of Claudin5 leads to changes in histamine-induced macromolecular leakage. In *Cldn5* iECKO mice, the heart vasculature still showed high resistance to histamine-mediated leakage, exhibiting an at least 100-fold lower dextran signal than other tissues (*Figure 3—figure supplement 1B*). The skeletal muscle vasculature also showed no significant change in permeability upon loss of Cldn5 (*Figure 3H*). In contrast, in the ear dermis, systemic administration of histamine resulted in a more than twofold increase in the extravasation of 2000 kDa dextran in *Cldn5* iECKO mice (*Figure 3I*). Given the increased sensitivity of the tracheal and back skin vasculature to histamine, analyses were repeated using a lower dose of histamine. Both back skin and trachea vasculatures showed a significant 1.5-fold increase in the leakage of 2000 kDa dextran in *Cldn5* iECKO mice compared to control (*Figure 3J and K*).

Oxazolone-induced dermatitis was employed as a model to assess the effect of chronic, endogenously produced inflammatory cytokines on the EC barrier. Following onset of dermal inflammation fluorescent tracers were administered and their subsequent extravasation analysed. This showed a significant increase in vascular leakage in the ear dermis (*Figure 3L*), but only a minor increase in the back skin, of *Cldn5* iECKO mice compared to control (*Figure 3M*), in keeping with a more limited role for Claudin5 in the back skin versus the ear skin (*Figure 3I and J*).

Claudin5 thus has a limited role in maintaining baseline EC barrier integrity in blood vessels outside of the CNS but is involved in the protection of vessels against agonist-induced macromolecular leakage in an organotypic manner.

## Loss of Claudin5 differentially affects vessel subtypes in the ear dermis

The increase in histamine-induced permeability in the ear dermis following the loss of Claudin5 prompted us to address in which vessel type the enhanced barrier breakdown is taking place. In the ear skin, expression of Claudin5 in arterioles was clearly reduced with some sporadic expression in *Cldn5* iECKO mice (*Figure 4A*). Even so, these vessels remained resistant to histamine-induced leakage. In an attempt to enhance the loss of Claudin5 without causing lethality, mice were treated topically on the ear skin with 4-hydroxytamoxifen (*Figure 4—figure supplement 1A*). Resultant loss of Claudin5 protein increased to approximately 85% (*Figure 4—figure supplement 1B*). The use of a *Rosa26* lox-STOP-lox-YFP reporter revealed high levels of recombination, however some remaining Claudin5 protein could be seen in arterioles and post-arteriolar capillaries (*Figure 4—figure supplement 1B*). Furthermore, these mice showed a similar increase in histamine-induced 2000 kDa dextran leakage as systemically tamoxifen treated mice and arterial ECs were resistant to leakage, even when devoid of apparent Claudin5 expression (*Figure 4—figure supplement 1C*). The lack of arteriolar leakage could potentially be explained by selective expression of histamine receptors on post-arteriolar ECs. However, *Hrh1* and *Hrh2*, the main histamine receptors believed to be involved in EC barrier disruption (*Adderley et al., 2015*; *Luo et al., 2013*), were generally either absent or expressed evenly between EC subtypes in all tissues (*Figure 4—figure supplement 1D*, *Figure 1—source data 1–4*). *Hrh1* and *Hrh2* did however show a slight increase in the venous populations in tracheal and skeletal muscle ECs capillary populations of skeletal muscle, respectively.

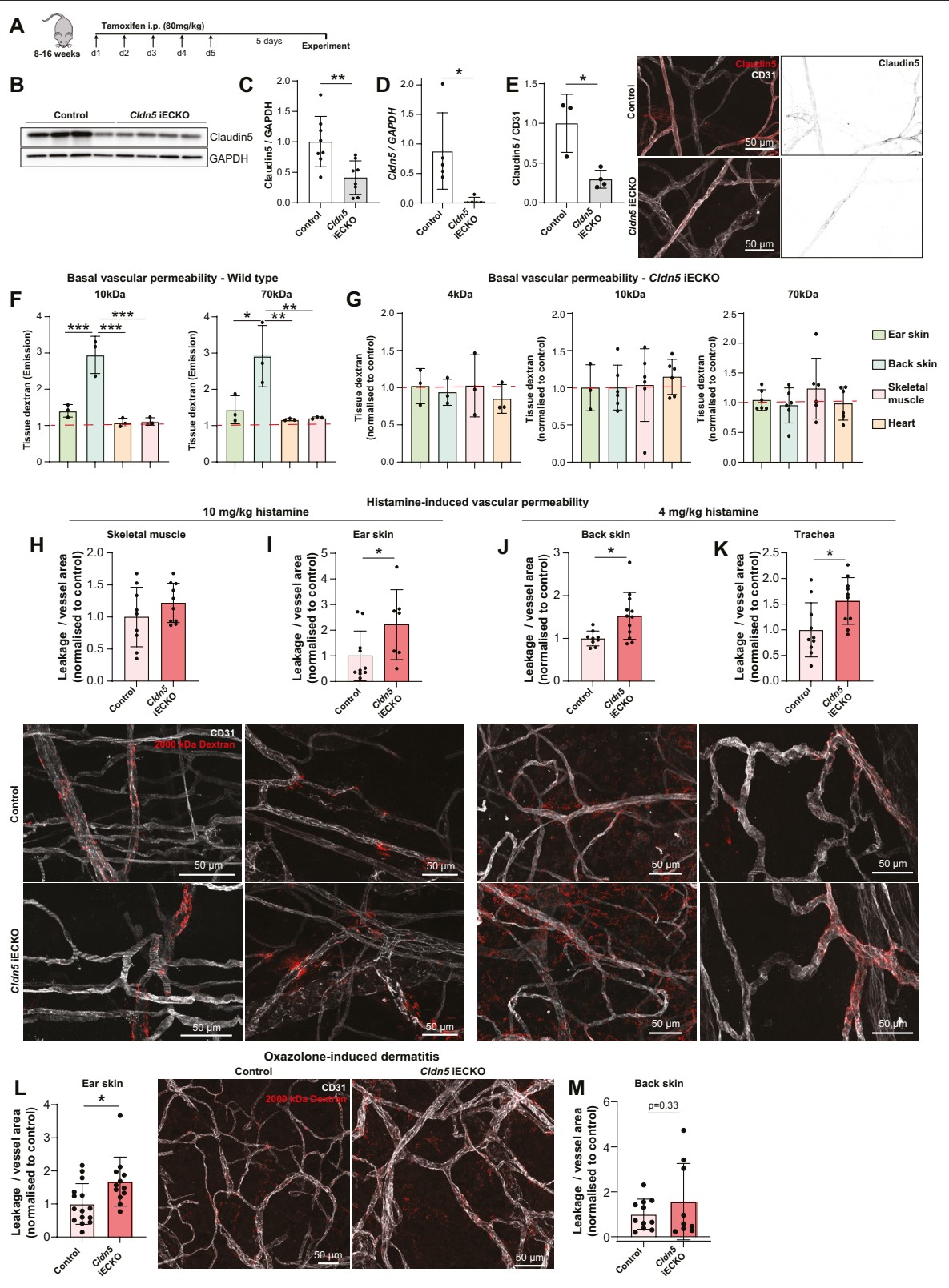

**Figure 3.** Claudin5 exhibits organotypic protection of the EC barrier. (**A**) Schematic illustration of systemic tamoxifen regime. (**B**) Representative western blot of Claudin5 protein expression in control and *Cldn5* iECKO mice. (**C**) Quantification of Claudin5 protein expression in lung lysates of control and *Cldn5* iECKO mice. n≥8 mice. (**D**) *Cldn5* gene expression by qPCR on lung lysates of control and *Cldn5* iECKO mice. n≥5 mice. (**E**) Claudin5 protein expression normalized to CD31 counter-staining in the ear skin of control and *Cldn5* iECKO mice following systemic tamoxifen. Right, representative

*Figure 3 continued on next page*

*Figure 3 continued*

images of Claudin5 immunostaining in control and *Cldn5* iECKO mice. n≥3 mice, 3 or more fields of view/mouse. (**F**) Blood vessel basal permeability to 10 kDa and 70 kDa dextran in ear skin, back skin, skeletal muscle, and heart of wildtype C57Bl/6 mice. Dashed lines represent background from control uninjected mice. n=3 mice. (**G**) Blood vessel basal permeability to 4 kDa, 10 kDa and 70 kDa dextran in ear skin, back skin, skeletal muscle, and heart of control and *Cldn5* iECKO mice. Dashed lines represent control Cre-negative mice. n≥3 mice. (**H–I**) Leakage of 2000 kDa dextran in response to systemic histamine stimulation (10 mg/kg) in skeletal muscle (**H**) and ear skin (**I**). Top, quantification of tracer leakage area / vessel area normalized to control (Cre-negative) mice. Bottom, representative images. n≥7 mice, 3 or more fields of view/mouse. (**J–K**) Leakage of 2000 kDa dextran in response to systemic histamine stimulation (4 mg/kg) in back skin (**J**) and trachea (**K**). Top, quantification of tracer leakage area / vessel area normalized to control (Cre-negative) mice. Bottom, representative images. n≥8 mice, 3 or more fields of view/mouse. (**L**) Quantification of 2000 kDa dextran leakage in the ear skin of control and *Cldn5* iECKO mice following Oxazolone-induced dermatitis. Right, representative images. n≥12 mice, 2 or more fields of view/mouse (**M**) Quantification of 2000 kDa dextran leakage in the back skin of control and *Cldn5* iECKO mice following Oxazolone-induced dermatitis. n≥9 mice, 2 or more fields of view/mouse Error bars; mean ± SD. Statistical significance: two-tailed paired Student's t test (**C-E**), H-M or one-way ANOVA with Tukey post-hoc test (multiple comparisons; **F–G**).

The online version of this article includes the following figure supplement(s) for figure 3:

**Figure supplement 1.** *Cldn5* targeting and histamine-induced leakage quantification.

Intravital visualisation of vascular leakage following intradermal histamine administration confirmed the EC barrier protective properties of Claudin5, with more leakage sites being induced per vessel length after histamine stimulation in the *Cldn5* iECKO ear dermis (*Figure 4B and C* and *Video 2*). The extent of barrier disruption at each site of leakage was also increased, with individual leakage sites exhibiting enhanced extravasation of dextran following Claudin5 loss (*Figure 4D*). The time lag of endothelial response to stimulation was however unchanged, with leakage occurring approximately 3 min after stimulation in *Cldn5* iECKO mice and their controls (*Figure 4E*).

We next explored in which vessel subtype leakage was enhanced following Claudin5 loss. Segregation of vessels into their subtypes in the dermis is, however, complicated by their stochastic organisation. Thus, determining precisely which vessels are affected by loss of Claudin5 is not possible. Post-arteriolar vessels were thus instead separated based on luminal dextran diameter. This analysis showed that small vessels (5–10 µm) and subsequent mid-sized venules (10–15 µm) leaked to a greater degree following the loss of Claudin5. In contrast, larger venules (15–20 µm) showed no change in leakage (*Figure 4F*). To explore further where these vessels lie within the vasculature, analysis of systemically-induced histamine leakage was carried out alongside visualisation of α-smooth muscle actin (αSMA), which is absent on capillaries but present on the arteriolar and venular aspect of the microvasculature. In the ear skin of *Cldn5*(BAC)-GFP mice, arteriolar αSMA coverage can be seen to end before *Cldn5*(BAC)-GFP expression, whilst its venular expression begins after *Cldn5*(BAC)-GFP expression has been lost (*Figure 4—figure supplement 1E*). *Cldn5*(BAC)-GFP expression is thus lost in the αSMA-negative capillary bed. Based on this segregation we observed an almost total loss of Claudin5 levels in capillaries and a 75% reduction in αSMA-positive arterioles (*Figure 4—figure supplement 1F*). Following histamine stimulation, analysis showed a 2.5-fold increase in leakage from αSMA-positive venules and a fourfold increase in leakage in αSMA-negative capillaries in *Cldn5* iECKO mice (*Figure 4G and H*). Furthermore, in *Cldn5* iECKO mice, leakage sites appeared closer to arterioles than in controls (*Figure 4I*). As expected, any dextran signal associated with arterioles was negligible and unaffected by the loss of Claudin5.

This data demonstrates that loss of Claudin5 decreases the junctional integrity of capillaries and venules, at least in the ear skin, and moves the limit between leakage resistant and susceptible ECs towards the arteriolar aspect. It is surprising that the phenotypic effects of *Cldn5* recombination is established in vessels seemingly negative for Cldn5 according to immunohistochemistry and the GFP reporter mouse. Use of more sensitive RNA in situ hybridisation, however, showed that in *Cldn5*(BAC)-GFP-negative vessels *Cldn5* expression still occurred, albeit to a considerably lower level than *Cldn5*(BAC)-GFP-positive vessels, in keeping with the expression patterning observed from scRNAseq data (*Figure 1F* and *Figure 4J*).

From the highly sensitive RNA in situ detection of *Cldn5* expression in capillaries and postcapillary venules, we conclude that in the wild-type vasculature, Claudin5 is responsible for limiting the disruption of EC junctions in capillaries, but also in immediately following venules. Arterioles remain resistant to agonist-induced leakage, as do larger venules, presumably due to residual Claudin5 expression and the relatively high expression of other TJ components.

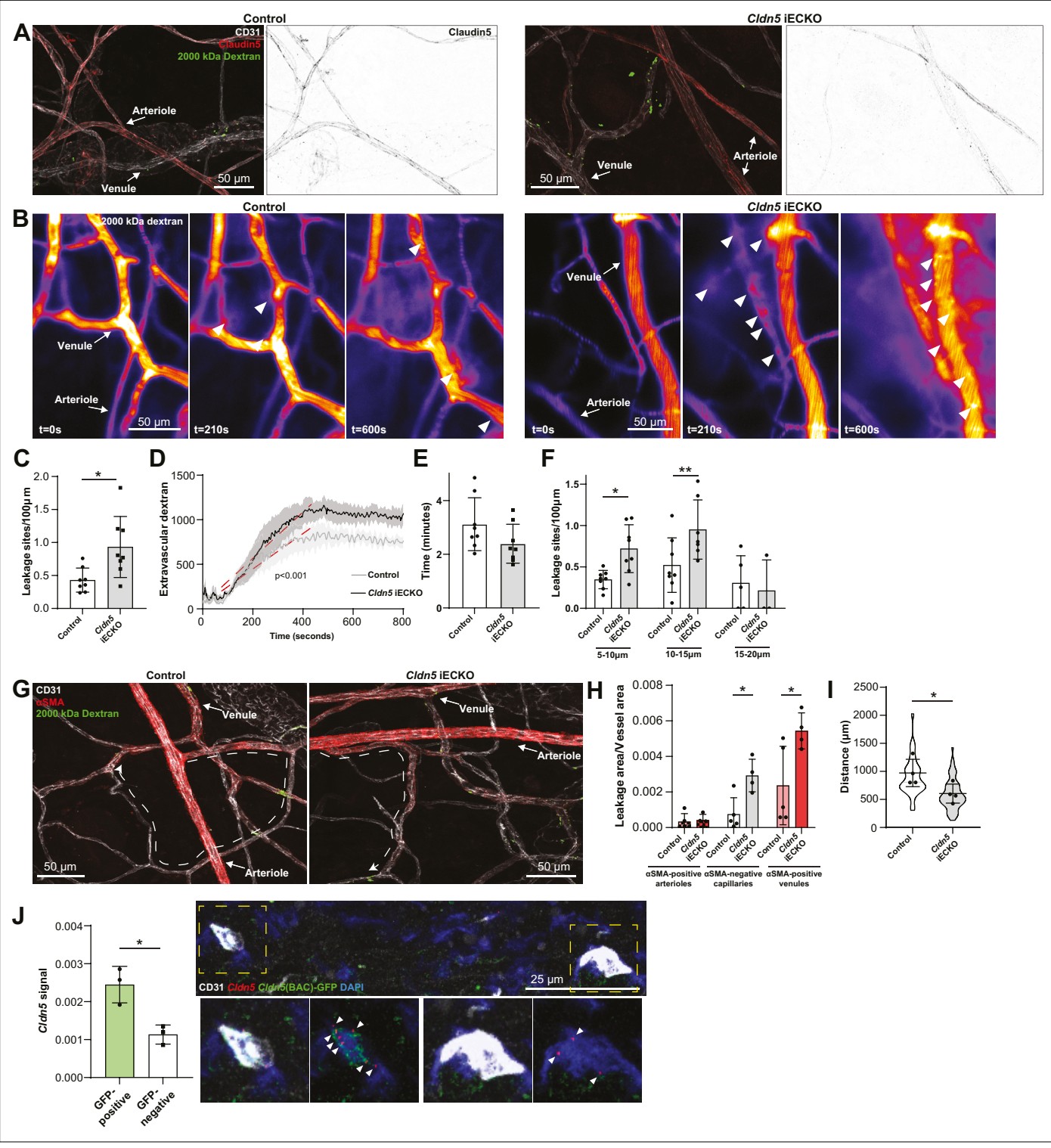

**Figure 4.** Loss of Claudin5 differentially affects vessel subtypes in the ear dermis. (**A**) Representative images of histamine-induced 2000 kDa dextran leakage in the ear skin of control (left) and *Cldn5* iECKO (right) mice. (**B**) Representative time-lapse images of 2000 kDa dextran leakage in response to intradermal histamine stimulation in the ear skin of control (left) and *Cldn5* iECKO (right) mice. Arrowheads show sites of leakage. (**C**) Leakage sites per vessel length in response to intradermal histamine stimulation in the ear skin of control and *Cldn5* iECKO mice. n≥7 mice, two or more acquisitions/mouse. (**D**) Quantification of extravascular 2000 kDa dextran over time in the ear skin of control and *Cldn5* iECKO mice following intradermal histamine stimulation. Red dashed lines represent lines of best fit for the slope between leakage initiation and leakage termination. n≥7 mice, two or more acquisitions/mouse. (**E**) Lag period between intradermal histamine injection and initiation of leakage in the ear skin of control and *Cldn5* iECKO mice.

*Figure 4 continued on next page*

**Figure 4 continued**

n≥7 mice, two or more acquisitions/mouse. (**F**) Leakage sites per length of post-arteriolar vessels of different diameter in response to intradermal histamine stimulation in the ear skin of control and *Cldn5* iECKO mice. n≥7 mice, two or more acquisitions/mouse. n≥7 mice, two or more acquisitions/mouse. (**G**) Representative images of 2000 kDa dextran leakage in response to systemic histamine stimulation in the ear skin of control and *Cldn5* iECKO mice counter-stained for αSMA. Dashed lines with arrows show distance from arteriolar/capillary transition to first site of leakage. (**H**) Leakage area/vessel area of 2000 kDa dextran in response to systemic histamine stimulation in αSMA-positive arterioles, αSMA-negative capillaries and αSMA-positive venules in the ear skin of control and *Cldn5* iECKO mice. n≥4 mice, 3 or more fields of view/mouse. (**I**) Distance between arteriolar-capillary branch points and the first site of 2000 kDa dextran leakage in response to systemic histamine stimulation in the ear skin of control and *Cldn5* iECKO mice. n≥4 mice, 3 or more fields of view/mouse. (**J**) *Cldn5* mRNA expression in *Cldn5*(BAC)-GFP-positive and -negative vessels of the ear skin. Left, quantification of *Cldn5* signal (*Cldn5* mRNA particles/vessel area). Right, representative image. Dashed boxes are magnified below, arrowheads mark *Cldn5* mRNA particles. n=3 mice, 4 or more fields of view/mouse. Error bars; mean ± SD. Statistical significance: two-tailed paired Student's t test (**C, E–J**) and linear regression and ANCOVA (**D**).

The online version of this article includes the following figure supplement(s) for figure 4:

**Figure supplement 1.** 4-hydroxytamoxifen-mediated Claudin5 loss, histamine receptor expression and vessel-specific leakage in *Cldn5* iECKO mice.

## Claudin5 modulates junction protein expression

The consequence of Claudin5 loss was further studied by transmission electron microscopy (TEM), with focus on non-arteriolar vessels in the ear dermis. In TEM images, the intercellular cleft is lined by parallel plasma membranes of contacting ECs and junctional complexes appear as electron dense structures following uranyl acetate staining. Analysis of the electron dense area, width and density showed that there was no obvious change in this junction structure after loss of Claudin5 (*Figure 5A–C*). In keeping with the enhanced leakage that we observed following the loss of Claudin5, greater disruption of the endothelial barrier was seen following histamine stimulation, allowing greater penetrance of horse radish peroxidase (HRP) into the intercellular cleft (*Figure 5D*).

The lack of apparent change in junction structure, but change in junction integrity, is surprising but may be explained by compensatory changes in junction composition. It is known that Claudin5 expression, through transcriptional cross-talk, can be controlled through VE-Cadherin and JAM-A (*Taddei et al., 2008*; *Kakogiannos et al., 2020*). Whether Claudin5 expression reciprocally controls the expression of other junctional proteins is unknown. We therefore investigated the expression of cell-cell adhesion proteins in *Cldn5* iECKO mice. Initially, lung RNA was screened for the expression of the AJ gene *Cdh5* (VE-Cadherin) and common TJ genes *Tjp1* (ZO-1), *Ocln* (Occludin), *F11r* (JAM-A), *Cgn* (Cingulin), and *Esam* (ESAM). Following *Cldn5* KO, small increases in *Cdh5*, *Ocln* and *F11r* expression, but a decrease in *Tjp1* expression was observed (*Figure 5E*). Subsequent analysis of lung protein samples confirmed the enhanced protein expression of VE-Cadherin and Occludin and decrease in ZO-1 (*Figure 5F*, *Figure 5—figure supplement 1A*). Correlation analysis of these samples also supported this finding, with Claudin5 expression levels inversely correlating with VE-Cadherin and Occludin and positively correlating with ZO-1, but not JAM-A, Cingulin, or ESAM (*Figure 5—figure supplement 1B*).

Immunohistochemistry analysis of the ear dermis similarly showed a downregulation of ZO-1 and an upregulation of VE-Cadherin in *Cldn5* iECKO mice (*Figure 5G and H*). Analysis of Occludin in these samples was precluded by its lack of expression in this vascular bed (*Figure 1E*). In contrast, ZO-1 and VE-Cadherin levels were unchanged in ECs of the back skin of *Cldn5* iECKO compared to control (*Figure 5I*). In the ear skin, VE-Cadherin and ZO-1 expression was further investigated to determine in which vessel types, according to αSMA expression, their change in expression is occurring. ZO-1 expression was found to be significantly decreased specifically in

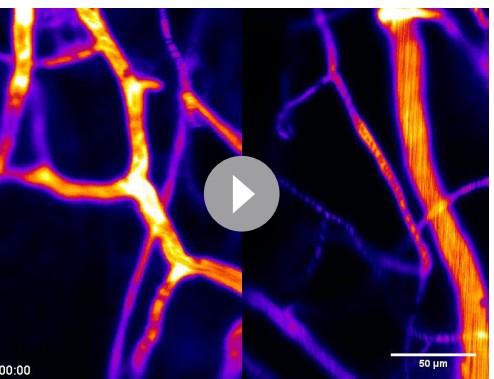

**Video 2.** Histamine-mediated leakage in control and *Cldn5* iECKO mice. Extravasation of circulating 2000 kDa FITC Dextran (pseudocolor) in control (left) and *Cldn5* iECKO (right) mice after intradermal injection of histamine in the ear dermis.

https://elifesciences.org/articles/78517/figures#video2

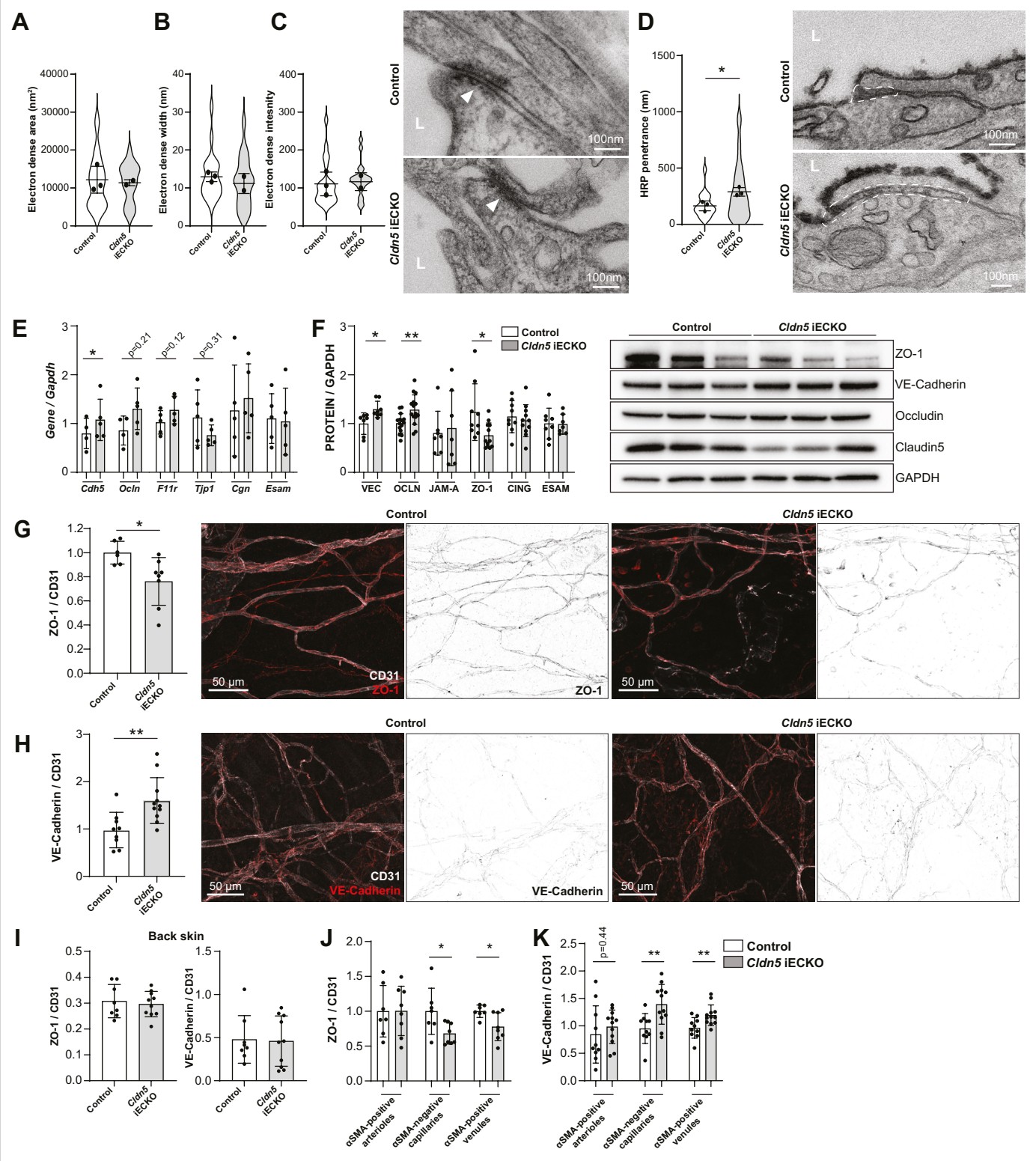

**Figure 5.** Claudin5 regulates junction protein expression. (**A–C**) Area (**A**) width (**B**) and intensity (**C**) of electron dense regions in the ear skin of control and *Cldn5* iECKO mice after visualisation by TEM. Right, representative TEM images of junctions in the ear skin of control and *Cldn5* iECKO mice. Junctions can be seen within electron dense regions (arrowheads). L, lumen. n≥2 mice, 6 or more fields of view/mouse. (**D**) Distance of HRP penetrance into EC junctions in the ear skin of control and *Cldn5* iECKO mice following systemic histamine stimulation. Right, representative TEM images of HRP penetrance (visualized by electron dense 3,3'-Diaminobenzidine (DAB) reaction precipitate) into EC junctions in the ear skin of control and *Cldn5* iECKO

*Figure 5 continued on next page*

*Figure 5 continued*

mice following systemic histamine stimulation. Dashed regions show areas of disrupted junction into which HRP has penetrated. Note that the typical electron dense area is lacking due to absence of uranyl acetate staining. L, lumen. n≥2 mice, 6 or more fields of view/mouse. (**E**) Gene expression of AJ- and TJ-associated genes in lung lysates of control and *Cldn5* iECKO mice. n≥4 mice. (**F**) Expression of AJ- and TJ- associated proteins in lung lysates of control and *Cldn5* iECKO mice. Right, representative western blots of AJ- and TJ- associated proteins in lung lysates of control and *Cldn5* iECKO mice. n≥4 mice. (**G**) Expression of ZO-1 in ear skin blood vessels of control and *Cldn5* iECKO mice. Left, quantification of ZO-1. Right, representative images of ZO-1 in the ear skin of control and *Cldn5* iECKO mice. n≥6 mice, 3 or more fields of view/mouse. (**H**) Expression of VE-Cadherin in ear skin blood vessels of control and *Cldn5* iECKO mice. Left, quantification of VE-Cadherin. Right, representative images of VE-Cadherin in the ear skin of control and *Cldn5* iECKO mice. n≥9 mice, 3 or more fields of view/mouse. (**I**) Quantification of ZO-1 (left) and VE-Cadherin (right) in back skin blood vessels of control and *Cldn5* iECKO mice. n≥8 mice, 2 or more fields of view/mouse. (**J**) Quantification of ZO-1 in different vessel subtypes in the ear skin of control and *Cldn5* iECKO mice. n≥3 mice, 3 or more fields of view/mouse. (**K**) Quantification of VE-Cadherin in different vessel subtypes in the ear skin of control and *Cldn5* iECKO mice. n≥3 mice, 3 or more fields of view/mouse. Error bars; mean ± SD. Statistical significance: two-tailed paired Student's t test.

The online version of this article includes the following figure supplement(s) for figure 5:

**Figure supplement 1.** Endothelial junction protein expression in *Cldn5* iECKO mice.

venules whilst VE-Cadherin was equally upregulated in both capillaries and venules, but not in arterioles (*Figure 5J and K*).

Claudin5 expression thus regulates the expression and localisation of other EC junction components. Consequently, whilst Claudin5 loss results in no overt changes in junction structure, in a tissue-specific manner it alters the regulatable and dynamic nature of the remaining junctional complex leading to changes in barrier stability.

## Discussion

The EC barrier consists of a variety of transmembrane cell adhesion molecules, creating endothelial junctions of highly variable composition and strictness. However, the correlation between specific expression patterns of cell adhesion molecules and the barrier strictness have remained unclear. Here, we investigated the heterogeneous nature of the BEC barrier and uncovered its composition and integrity in non-CNS, continuous endothelia. Broadly, vessels in the ear skin, back skin, trachea, skeletal muscle, and the heart share a similar complement of AJ and TJ genes. Still, subtle variation in their relative distribution between different vessel subsets are often evident, as is variability in EC barrier integrity and response to stimulation in diverse tissues. Differential gene expression analysis revealed variable gene expression between vessel subsets within tissues, which is provided as a resource along with expression values of genes typically associated with junction regulation and integrity (*Figure 1— figure supplements 3 and 5*, *Figure 1—source data 1–5*). Generally, transmembrane TJ genes were found to be expressed to a greater degree in the arterial aspect of the microvasculature. In particular, *Cldn5* stands out in this analysis as it exhibited a large gradual decrease in expression from arterioles through capillaries to venules in all tissues analysed, including in the human dermis. Previously we have shown that, in the ear skin, Claudin5 expression inversely correlates with susceptibility to VEGF-A-induced leakage, with capillaries possessing a mixture of Claudin5-positive and -negative cells being responsive to leakage stimulation (*Honkura et al., 2018*). This correlation however does not extend to other vascular beds such as the back skin and trachea, in which the leakage permissive vasculature was shifted away from the capillary region toward the venous side. To fully address why Claudin5-negative capillaries in the trachea might be resistant to leakage requires more extensive insights into the junctional organisation in these organs. Of note, whilst tracheal capillaries might be resistant to leakage, the trachea shows a much larger leakage response overall than the ear skin. The back skin similarly showed a larger leakage response than the ear skin. These data reveal the highly heterogeneous nature of barrier integrity and patterning across different tissues.

Differential regulation of EC barrier integrity in tissue-specific vasculatures was also observed following removal of Claudin5 in adult mice. We find that outside of the CNS, the largest influence of Claudin5 is in the ear skin among the organs analysed, with a more modest loss in barrier integrity also observed in back skin and tracheal vasculatures. Similarly, a significant loss in barrier integrity was seen in the ear skin, but not the back skin, following the loss of Claudin5 in an Oxazolone-induced model of dermatitis. EC barrier integrity in skeletal muscle and heart meanwhile showed no change following the loss of Claudin5. This observation might be explained by the expression patterning of *Cldn5*,

which declines more proximal to the arterial side along the arteriovenous axis in skeletal muscle than tissue such as the ear skin (*Figure 1F*). Alternatively, we observe expression of other TJ genes, such as *Ocln*, *Nectin2* and *Nectin3*, in skeletal muscle and heart which are not evident in the other tissues analysed. Other studies have demonstrated a barrier protective role for Occludin, nectin-2 and nectin-3 in vitro (*Martin et al., 2013*; *Son et al., 2016*; *Murakami et al., 2009*), but further studies are required to establish whether these actively participate in EC barrier integrity in vivo (*Saitou et al., 2000*).

In keeping with previous observations in the brain, loss of Claudin5 did not alter BEC junction structure in the ear dermis (*Nitta et al., 2003*). Similarly, loss of VE-Cadherin in the lungs enhances barrier permeability without causing any structural defects (*Duong et al., 2020*). Interestingly, concomitant loss of ESAM along with VE-Cadherin produces more overt changes in junction structure and enhances loss of barrier integrity, highlighting the redundant organisation of the EC barrier (*Duong et al., 2020*). Furthermore, interpretation of models that manipulate the EC barrier are often complicated by compensatory effects in gene expression. For example, VE-cadherin has been found to be a major regulator of EC gene expression (*Morini et al., 2018*). Assembly of VE-Cadherin junctions upregulates Claudin5 expression, whilst its loss leads to compensatory upregulation of N-Cadherin (*Taddei et al., 2008*; *Giampietro et al., 2012*). We find here in vivo that Claudin5 can reciprocally alter expression of VE-Cadherin, as well as Occludin and ZO-1 (*Figure 5E and F*). Tight junctions have previously been suggested to control gene expression through ZO-1's tension-dependent interaction with the ZO-1–associated nucleic acid binding protein (ZONAB) (*Balda and Matter, 2000*; *Spadaro et al., 2017*). When under tension ZO-1 can interact with ZONAB, resulting its junctional sequestering. ZO-1 also interacts with Claudins through their PDZ domains (*Itoh et al., 1999*). Loss of junctional Claudin5 and resulting dislocation of junctional ZONAB may thus modify the EC transcriptome. Upregulation of VE-Cadherin meanwhile may explain the lack of structural defects of junctions in *Cldn5* iECKO mice. Moreover, in contrast to Claudin5, VE-Cadherin at junctions is highly dynamic and its localisation is regulated by numerous cytokines and growth factors that affect its phosphorylation status (*Smith et al., 2020*; *Orsenigo et al., 2012*; *Eliceiri et al., 1999*). Upregulation of VE-Cadherin by Claudin5 deficiency is therefore likely to produce junctions that are less stable and subject to higher turnover in a stimulatory environment.

Collectively, the data in this study highlights the variable regulation and integrity of the EC barrier in various vascular beds and demonstrates a distinct role for Claudin5 in the EC barrier integrity of different tissues. The mechanisms underlying such organotypic barrier integrity and differential patterning of junctional components are currently poorly understood. Pericytes are known to be essential for maintenance of the tight BBB (*Armulik et al., 2010*). Outside of the CNS however pericyte coverage is comparatively low and whether coverage differs between non-CNS vascular beds is unknown. Barrier integrity may also be regulated by basement membrane components such as laminin α5, which enhances VE-Cadherin stability at cell-cell junctions (*Richards et al., 2021*; *Song et al., 2017*). Similar to pericytes, basement membrane coverage differs between different vascular beds, as well as between vessel subtypes (*Richards et al., 2021*; *Di Russo et al., 2017*). Whether basement membrane components significantly define EC junction composition however is currently unknown. A better understanding of basement membrane, as well as pericyte, coverage however would provide potential new means of manipulating EC biology for therapeutic benefit.

In conclusion, this study provides an in-depth characterisation and comparison of the EC barrier in numerous organs. Furthermore, in accordance with its role in forming the tight BBB and BRB in the CNS, we find that Claudin5 influences vascular permeability in peripheral vascular beds, albeit in an organotypic and vessel-type manner. The impact of Claudin5 in organ-specific vascular permeability was demonstrated by the direct correlation between its expression and high barrier integrity in some, but not all vascular beds. Variability in the pattering of the EC barrier between vascular beds and vessel subtypes, as well as the redundancy, interdependency and crosstalk between different junction components have remained poorly defined. Understanding the role of different EC components in various vascular beds will allow us to better appreciate organ-specific defects in vascular permeability and how they may be therapeutically targeted.

# Materials and methods

**Key resources table**

| Reagent type (species) or resource | Designation | Source or reference | Identifiers | Additional information |
|---|---|---|---|---|
| Genetic reagent (*Mus musculus*) | C57BL/6 J | Taconic | B6-F/M | |
| Genetic reagent (*Mus musculus*) | *Cldn5(BAC)-GFP* | *Laviña et al., 2018* | N/A | |
| Genetic reagent (*Mus musculus*) | *Cldn5 iECKO* | This paper | N/A | See *Figure 3—figure supplement 1* and Methods-Animals |
| Genetic reagent (*Mus musculus*) | *Cldn5$^{fl/fl}$; Rosa26$^{lox-STOP-lox-YFP}$; Cdh5$^{CreERT2}$* | This paper | N/A | See Methods-Animals |
| Antibody | Mouse monoclonal anti-GAPDH | Millipore | MAB374 | 1:1,000 |
| Antibody | Rat monoclonal anti-CD31 | BD Biosciences | 553370 | 1:100 |
| Antibody | Goat polyclonal anti-CD31 | R&D Systems | AF3628 | 1:100 |
| Antibody | Goat polyclonal anti-VE-Cadherin | R&D Systems | AF1002 | 1:100, 1:1000 |
| Antibody | Chicken polyclonal anti-GFP | Abcam | Ab13970 | 1:100 |
| Antibody | Rabbit polyclonal anti-Claudin5 | ThermoFischer Scientific | 341600 | 1:100, 1:1000 |
| Antibody | Rabbit polyclonal anti-ZO-1 | ThermoFischer Scientific | 617300 | 1:100, 1:1000 |
| Antibody | Rabbit polyclonal anti-Occludin | ThermoFischer Scientific | 711500 | 1:1,000 |
| Antibody | Rabbit monoclonal anti-JAM-A | *Martìn-Padura et al., 1998* | N/A | 1:1000 |
| Antibody | Rabbit polyclonal anti-Cingulin | *Cardellini et al., 1996* | N/A | 1:1000 |
| Antibody | Goat polyclonal anti-ESAM | R&D Systems | AF2827 | 1:1000 |
| Antibody | Goat polyclonal anti-collagen IV | Merck Millipore | AB789 | 1:100 |
| Antibody | Mouse monoclonal anti-αSMA FITC | Sigma Aldrich | F3777 | 1:100 |
| Antibody | Mouse monoclonal anti- αSMA Cy3 | Sigma Aldrich | C6198 | 1:100 |
| Antibody | Rat monoclonal Anti-CD16/32 | ThermoFischer Scientific | 14-0161-85 | 1:100 |
| Antibody | Rat monoclonal Anti-CD31 FITC | BD Biosciences | 553372 | 1:50 |
| Antibody | Rat monoclonal Anti-CD45 APC | BioLegend | 103112 | 1:50 |
| Antibody | Rat monoclonal Anti-Lyve1 eFluor 660 | ThermoFischer Scientific | 50–0443082 | 1:50 |
| Antibody | Donkey polyclonal anti-rat alexa 488 | ThermoFischer Scientific | A21208 | 1:400 |
| Antibody | Donkey polyclonal anti-rat alexa 594 | ThermoFischer Scientific | A21209 | 1:400 |
| Antibody | Donkey polyclonal anti-rabbit alexa 488 | ThermoFischer Scientific | A21206 | 1:400 |
| Antibody | Donkey polyclonal anti-rabbit alexa 568 | ThermoFischer Scientific | A10042 | 1:400 |

*Continued on next page*

*Continued*

| Reagent type (species) or resource | Designation | Source or reference | Identifiers | Additional information |
|---|---|---|---|---|
| Antibody | Donkey polyclonal anti-goat alexa 647 | ImmunoResearch Laboratories | 705-605-147 | 1:400 |
| Antibody | Donkey polyclonal anti-chicken alexa 488 | ImmunoResearch Laboratories | 703-545-155 | 1:400 |
| Antibody | Sheep polyclonal anti-mouse HRP | Cytiva | NA931 | 1:10,000 |
| Antibody | Sheep polyclonal anti-rabbit HRP | Cytiva | NA934 | 1:10,000 |
| Sequence-based reagent | *Cldn5* probe | ACD Bio | 491611-C2 | |
| Sequence-based reagent | 3-plex negative control probes | ACD Bio | 320871 | |
| Sequence-based reagent | 3-plex positive control probes | ACD Bio | 320811 | |
| Sequence-based reagent | *GAPDH* | ThermoFischer Scientific | Mm99999915_g1 | |
| Sequence-based reagent | *Cldn5* | ThermoFischer Scientific | Mm00727012_s1 | |
| Sequence-based reagent | *Cdh5* | ThermoFischer Scientific | Mm00486938_m1 | |
| Sequence-based reagent | *Tjp1* | ThermoFischer Scientific | Mm01320638_m1 | |
| Sequence-based reagent | *Ocln* | ThermoFischer Scientific | Mm00500912_m1 | |
| Sequence-based reagent | *F11r* | ThermoFischer Scientific | Mm00554113_m1 | |
| Sequence-based reagent | *Cgn* | ThermoFischer Scientific | Mm01263534_m1 | |
| Sequence-based reagent | *Esam* | ThermoFischer Scientific | Mm00518378_m1 | |
| Peptide, recombinant protein | Collagenase IV | Worthington | LS004183 | |
| Peptide, recombinant protein | DNase I | Worthington | LS006333 | |
| Peptide, recombinant protein | HRP | SigmaAldrich | 77332 | |
| Commercial assay or kit | RNAscope Fluorescent Multiplex Assay | ACD Bio | 322340, 320851 | |
| Commercial assay or kit | RNeasy Plus kit | Qiagen | 74034 | |
| Commercial assay or kit | iScript Adv cDNA Kit for RT-qPCR | Bio-Rad | 1725038 | |
| Chemical compound, drug | Tamoxifen | SigmaAldrich | T5648 | |
| Chemical compound, drug | 4-hydroxytamoxifen | SigmaAldrich | H7904 | |
| Chemical compound, drug | Oxazolone | SigmaAldrich | E0753 | |
| Chemical compound, drug | Histamine | SigmaAldrich | H7125 | |
| Other | Live/Dead near IR cell stain | ThermoFischer Scientific | L10119 | See Materials and methods-Ear dermal single cell isolation |
| Other | Phosphatase inhibitor cocktail | Roche | 04906837001 | See Materials and methods-Western blot analysis |
| Other | LDS sample buffer | Invitrogen | NP0007 | See Materials and methods-Western blot analysis |

*Continued on next page*

*Continued*

| Reagent type (species) or resource | Designation | Source or reference | Identifiers | Additional information |
|---|---|---|---|---|
| Other | Sample reducing agent | Invitrogen | NP0009 | See Materials and methods-Western blot analysis |
| Other | MOPS SDS running buffer | Invitrogen | NP0001 | See Materials and methods-Western blot analysis |
| Other | PVDF membrane | Thermofischer Scientific | 88518 | See Materials and methods-Western blot analysis |
| Other | NuPAGE transfer buffer | Novex | NP006 | See Materials and methods-Western blot analysis |
| Other | RNAlater | ThermoFischer Scientific | AM7024 | See Materials and methods-quantitative PCR |
| Other | 2000 kDa FITC Dextran | SigmaAldrich | FD2000S | See Materials and methods-permeability analysis |
| Other | 2000 kDa TRITC Dextran Lysine Fixable | ThermoFischer Scientific | D7139 | See Materials and methods-permeability analysis |
| Other | 10 kDa TRITC Dextran | ThermoFischer Scientific | D1817 | See Materials and methods-permeability analysis |
| Other | 4 kDa TRITC Dextran | Tdb labs | TD4 | See Materials and methods-permeability analysis |
| Other | 10 kDa FITC Dextran | Tdb labs | FD10 | See Materials and methods-permeability analysis |
| Other | 70 kDa TRITC Dextran | Tdb labs | TD70 | See Materials and methods-permeability analysis |
| Other | 2000 kDa FITC Dextran lysine fixable | Tdb labs | FLD2000 | See Materials and methods-permeability analysis |
| Other | Anti-Isolectin GS-IB4 | Molecular Probes | I32450 | See Materials and methods-Immunohistochemistry |

## Animals

For generation of a mouse ear dermal scRNAseq dataset, wild-type female C57BL/6 J mice aged 8–16 weeks were used. *Cldn5*<sup>flox/flox</sup> were generated by Taconic by flanking the sole exon with *loxP* sites, introduced by homologous recombination on the genetic C57Bl/6 black background (*Figure 3—figure supplement 1A*). This strain was crossed with Cdh5(PAC)-CreER$^{T2}$ mice (kind gift from Dr. Ralf Adam, Max-Planck Institute Münster) to generate endothelial specific knockout of *Cldn5*. *Cldn5*(BAC)-GFP mice have been described previously (*Honkura et al., 2018*; *Laviña et al., 2018*). To monitor Cre recombinase activity with YFP expression, the fluorescent reporter mouse line B6.129 × 1-Gt(ROSA)26Sor<sup>tm1(EYFP)Cos</sup>/J (Stock Number 006148, The Jackson Laboratory) was introduced. Both males and females, aged 8–18 weeks, were included in experiments. In vivo animal experiments were carried out in accordance with the ethical permit provided by the Committee on the Ethics of Animal Experiments of the University of Uppsala (permit 6789/18). Mice were maintained in ventilated cages with group housing (2–5 per cage) and access to water and feed ad libitum. Each experiment was conducted on tissue from at least three age-matched animals representing individual biological repeats. Sample size (number of acquired images / movies and number of mice) were chosen to ensure reproducibility and allow stringent statistical analysis. To induce Cre recombinase-mediated gene recombination tamoxifen (SigmaAldrich, T5648) was injected intraperitoneally (1 mg/day) for 5 consecutive days. The mice were allowed to rest for 5 days before experiments were conducted. For topical tamoxifen induction 50 µg of 4-hydroxytamoxifen (SigmaAldrich, H7904) in acetone was applied to each side of the mouse ear for 3 consecutive days twice. Experiments were conducted 30 days after initial treatment (*Figure 4—figure supplement 1A*).

## Intravital vascular leakage assay

Intravital imaging of the mouse ear with intradermal injection has been described previously (*Honkura et al., 2018*). Briefly, following systemic administration of 2000 kDa FITC (SigmaAldrich, FD2000S) or

TRITC Dextran (ThermoFischer Scientific, D7139) by tail-vein injection, mice were sedated by intraperitoneal injection of Ketamine-Xylazine (120 mg/kg Ketamine, 10 mg/kg Xylazine) and the ear secured to a solid support. Mice were maintained at a body temperature of 37 °C for the entire experiment, maximum 90 min. Time-lapse imaging was performed using single-photon microscopy (Leica SP8). For intradermal EC stimulation, a volume of approximately 0.1 µl histamine (SigmaAldrich, H7125), concentration 10 ng/µl, was injected using a sub-micrometer capillary needle. 10 kDa TRITC Dextran (ThermoFischer Scientific, D1817) was used as a tracer. Leakage sites were identified in time-lapse imaging as defined sites of concentrated dextran in the extravascular space.

## Permeability analysis

To assess EC permeability, all tissues were cleaned of excess cartilage, fat and connective tissue. Skeletal muscle was assessed using the tibialis anterior.

To analyse baseline permeability mixtures of dextran (Tdb labs; 4 kDa, TD4, 10 kDa, FD10, 70 kDa, TD70) (20 g/kg) were injected systemically through the tail vein. Four hours later mice were anesthetized with Ketamine/Xylazine before intracardiac perfusion with Dulbecco's phosphate buffered saline (DPBS). Tissues were dissected, washed in DPBS and incubated in formamide for 48 hr at 55 °C. Dextran fluorescence was then measured using a spectrophotometer and normalized to tissue weight.

To assess histamine induced leakage microscopically, mixtures of 2000 kDa lysine fixable dextran (FITC, Tdb labs; TRITC, ThermoFischer Scientific, D7139) (20 g/kg) with histamine (4 or 10 mg/kg) were injected systemically through the tail vein. Ten minutes later, mice were anesthetized with Ketamine/Xylazine before intracardiac perfusion with DPBS followed by 1% paraformaldehyde. Tissues were then immersed in 1% paraformaldehyde for 2 hr before proceeding with immunohistochemistry. For leakage quantification at least three large tile scan areas ($\geq 1$ mm$^2$) were captured for each mouse.

## Oxazolone-induced dermatitis

Mice were pre-sensitized using 50 µl of 2% Oxazolone in acetone to the belly. After 5 days, mice were treated with 2% Oxazolone (10 µl to the ear, 50 µl to the shaved back skin). Twenty-four hr later mice were injected systemically through the tail vein with a mixture of 2000 kDa lysine fixable dextran (TRITC, ThermoFischer Scientific, D7139) (20 g/kg). Tracers were allowed to circulate for 1 hr before mice were anesthetized with Ketamine/Xylazine and intracardiac perfusion with DPBS followed by 1% paraformaldehyde was carried out. Tissues were then immersed in 1% paraformaldehyde for 2 hr before proceeding with immunohistochemistry.

## Immunohistochemistry

Tissues were fixed through intracardiac perfusion with 1% paraformaldehyde (PFA) followed by immersion in 1% PFA for 2 hr at room temperature. For whole mount preparation of the ear skin, back skin and trachea removal of excess cartilage, fat and connective tissue tissues was carried out. Tissues were blocked overnight at 4 °C in Tris-buffered saline (TBS), 5% (w/v) Bovine Serum Albumin (BSA), 0.2% Triton X-100. Samples were incubated overnight with primary antibody in blocking solution, followed by washing in TBS, 0.2% Triton X-100 and incubation with appropriate secondary antibody for 2 hr at room temperature in blocking buffer before washing and mounting in fluorescent mounting medium (DAKO).

For assessment of skeletal muscle (tibialis anterior) and heart tissues were equilibrated in 30% sucrose before being embedded in optimal cutting temperature (OCT). Sections consisted of 100 µm vibratome sections covering the whole tibialis anterior along its long axis and transverse sections of the left ventricle of the heart. Sections were blocked for 30 min at room temperature in Tris-buffered saline (TBS), 5% (w/v) Bovine Serum Albumin (BSA), 0.2% Triton X-100. Samples were incubated for 2 hr with primary antibody in blocking solution, followed by washing in TBS, 0.2% Triton X-100 and incubation with appropriate secondary antibody for 1 hr at room temperature in blocking buffer before washing and mounting in fluorescent mounting medium (DAKO). Images were acquired using a Leica SP8 confocal microscope.

Commercial antibodies used were: rat anti-CD31 (BD Biosciences, 553370), goat anti-CD31 (R&D Systems, AF3628), goat anti-VE-Cadherin (R&D Systems, AF1002), chicken anti-GFP (Abcam, ab13970) (1:400), rabbit anti-Claudin5 (ThermoFischer Scientific, 341600), rabbit anti-ZO-1 (ThermoFischer Scientific, 617300), goat anti-Collagen IV (Merck Millipore, AB789), mouse anti-αSMA FITC (Sigma

Aldrich, F3777), mouse anti-αSMA Cy3 (Sigma Aldrich, C6198), *Griffonia simplicifolia* Isolectin GS-IB4 (Molecular Probes, I32450) (1:400). Secondary antibodies against rat (ThermoFischer Scientific; Alexa 488, A21208 and Alexa 594, A21209), rabbit (ThermoFischer Scientific; Alexa 488, A21206 and Alexa 568, A10042), goat (ImmunoResearch Laboratories, Alexa 647, 705-605-147) and chicken (ImmunoResearch Laboratories, Alexa 488, 703-545-155) were used. Primary and secondary antibodies were prepared at a dilution of 1:100 and 1:400, respectively unless otherwise stated.

## In situ RNA hybridisation

Detection of *Cldn5* mRNA in the ear dermis was performed using RNAscope Fluorescent Multiplex Assay (ACD Bio, 322,340 and 320851) according to the manufacturer's instructions (https://www.acdbio.com). Briefly, 10 μm transverse cryosections of the ear dermis were fixed in chilled 4% PFA for 15 minutes (4 °C) then dehydrated by incubating in increasing concentrations of ethanol (50%, 75%, 100%, 100% for 5 minutes each). Following protease IV digest (30 min at RT), Cldn5 probe (ACD Bio, 491611-C2) was hybridized on the tissue sections for 2 hr at 40 °C. 3-plex negative (ACD Bio, 320871) and positive (ACD Bio, 320811) controls were used to confirm signal specificity. Signal amplification and detection was conducted using reagents included in the Fluorescent Multiplex Reagent Kit. Following the RNAscope procedure, sections were immediately counterstained with Hoechst (Molecular Probes, H3570) (1:1000), *Griffonia simplicifolia* Isolectin GS-IB4 (Molecular Probes, I32450) and αSMA (Sigma Aldrich, F3777) (1:300) for 1 hr at RT to visualize nuclei, blood vessels, and mural cells, respectively. Images were acquired using a Leica SP8 confocal microscope with at least three sections analysed / mouse. Fluorescent dots representing one mRNA molecule were quantified in *Cldn5*(BAC)-GFP-negative and -positive vessels before normalising to these vessel areas. Images were acquired using a Leica SP8 confocal microscope with at least 3 sections analysed / mouse. Fluorescent dots representing one mRNA molecule were quantified in *Cldn5*(BAC)-GFP-negative and -positive vessels before normalising to the respective vessel areas.

## Ear dermal single-cell isolation

Following cervical dislocation, ears of wild-type C57BL/6 J mice were collected and mechanically disrupted before enzymatic dissociation in 10 mg/ml Collagenase IV (Worthington, LS004183), 0.2% FBS and 0.2 mg/ml DNase I (Worthington, LS006333) in PBS at 37 °C for 20 min. Following removal of debris using a 50 μm filter, cells were resuspended in FACS buffer (0.5% FBS, 2 mM EDTA in PBS) and incubated with an anti-CD16/32 antibody (ThermoFischer Scientific, 14-0161-85) on ice for 10 min. Subsequently, cells were incubated with antibodies against CD31-FITC (BD Biosciences, 553372), CD45-APC (BioLegend, 103112) and Lyve1-eFluor 660 (ThermoFischer Scientific, 50-0443-82) for 30 min before washing and addition of live/Dead near IR cell stain (ThermoFischer Scientific, L10119) to distinguish viable cells. CD31+/CD45-/Lyve1- cells were obtained by FACS following filtration through a 50 μm mesh using a BD FACSAria III (100 μm nozzle size, 20 psi sheet pressure) at the BioVis core facility at the Department of Immunology, Genetics and Pathology (IGP), Uppsala University. Cells were captured directly into 2.3 μl lysis buffer (0.2% Triton-X Sigma, cat: T9284), 2 U/μl RNase inhibitor (ClonTech, cat: 2,313B), 2 mM dNTP's (ThermoFisher Scientific, cat: R1122), 1 μM Smart-dT30VN (Sigma), ERCC 1:$4 \times 10^7$ dilution (Ambion, cat: 4456740) in 384-well plates prior to library preparation.

## Smart-seq2 library preparation and sequencing

Single-cell libraries were prepared as described previously (*Picelli et al., 2014*), with the following specifications: 0.0025 μl of a 1:40,000 diluted ERCC spike-in concentration stock and all cDNA was amplified with 22 PCR cycles before QC control with a Bioanalyzer (Agilent Biosystems). The libraries were sequenced on a HiSeq2500 at the National Genomics Infrastructure (NGI), Science for Life Laboratory, Sweden, with single 50 bp reads (dual indexing reads). All single-cell transcriptome data were generated at the Eukaryotic Single-cell Genomics facility at Science for Life Laboratory in Stockholm, Sweden.

## Data processing

Raw data for ear skin sequencing data was aligned to the mouse reference genome mm10 with tophat (v.2.1.1) (*Kim et al., 2013*), duplicated reads were filtered out using the samtools software

(v.0.1.18), and gene counts were summarized using *featureCounts* function from the Subread package (v.1.4.6-p5) (*Liao et al., 2014*). Raw counts of trachea and heart BEC generated by the Tabula Muris Consortium (*Tabula Muris Consortium, 2020*) were obtained from https://s3.console.aws.amazon.com/s3/buckets/czb-tabula-muris-senis/. Both FACS (trachea) and droplet (trachea and muscle) processed data was used. Previously published heart and skeletal muscle data (*Kalucka et al., 2020*) was obtained from ArrayExpress (https://www.ebi.ac.uk/arrayexpress/), accession number E-MTAB-8077, and preprocessed with cellranger (v3.0.2) using the mm10 reference genome. Human dermal BEC data (*Li et al., 2021*) was obtained from the National Genomics Data Center (https://bigd.big.ac.cn/), accession number PRJCA002692, and preprocessed with cellranger (v5.0.1) using the GRCh38 reference genome.

Blood vessel ECs were defined by their expression of *Pecam1*, *Cdh5*, and *Kdr*. Non-BEC (defined by *Lyve1*, *Prox1*, *Ptprc*, *Pdgfrb*, or *Kcnj8* expression) and cells with a total count below 600, fewer than 500 or more than 8,000 expressed genes, were removed from downstream analysis. A total of 534 (356+178) ear skin, 559 (356+203) trachea, 3,498 (1118+2380) skeletal muscle, and 6,423 (2923+3500) heart BEC were included in the analysis. 8,518 BEC were identified and included in the human dermal dataset.

All processing steps were carried out with R (v.4.0.4; *Lost Library Book*) unless stated otherwise. Size factor-based normalization (*McCarthy et al., 2017*) and log transformation was carried out with scran (v.1.18.7) *quickCluster* and *computeSumFactors,* followed by batchelor (v.1.6.3) *multiBatchNorm* in order to adjust for differences in sequencing depth between samples and individual cells; scuttle (v.1.0.4) *logNormCounts* was used when preparing the human dermal BEC data. Highly variable genes were identified with scran *modelGeneVar* using non-integrated data while blocking for sample specific bias. Imputation was done with the magic python package (v.0.1.1; k=9, ka = 3, t=1, 2, 4, or 6) in order to reduce the effects of dropouts, and integration of the datasets was done with batchelor *fastMNN* (k=100) (*van Dijk et al., 2018*; *Haghverdi et al., 2018*). Trajectory inference was done using tSpace (v.0.1.0) and a total of 400 trajectories were calculated using imputed (t=6) and integrated expression of the 1,000 most variable genes identified in the murine data, or the imputed (t=6) human data (*Dermadi et al., 2020*). The average of two trajectories in the murine datasets, and one trajectory in the human dermal dataset, identified as spanning from arterial to venous BEC was separated into equidistant bins and the mean gene expression was calculated for each bin and for each organ. The bins were subsequently annotated based on the mean expression as belonging to the subtypes described in the main text. UMAP calculations were done with umap (v.0.2.7.0) using imputed expression of the 1,000 most variable genes. Heatmaps were generated using ComplexHeatmap (v.2.6.2), and violin plots were generated using ggplot2 (v.3.3.5) or GraphPad Prism (v.9.0.0). Both plot types display imputed gene expression (t=2, murine data; t=1, human data) without MNN-integration. Differential gene expression analysis comparing one subtype to all other cells in a sample was carried out using Seurat (v.4.0.5) *FindConservedMarkers* (mouse datasets) or *FindMarkers* (human dataset).

## Western blot analysis

Lungs from control and *Cldn5* iECKO mice were removed and snap frozen. Protein was obtained by mechanical dissociation in RIPA buffer supplemented with 50 nM Na3VO4, Phosphatase inhibitor cocktail (Roche 04906837001) and Protease inhibitor cocktail (Roche, 04693116001). LDS sample buffer (Invitrogen, NP0007) and Sample Reducing Agent (Invitrogen, NP0009) were added to the samples and heated to 70 °C for 10 min. Proteins were separated on Nu Page 4–12% Bis-Tris Gel (Invitrogen) in MOPS SDS Running buffer (Invitrogen, NP0001), transferred to PVDF membrane (Thermo scientific, 88518) in NuPAGE transfer buffer (Novex, NP006), 10% methanol and subsequently blocked with 5% BSA in Tris-buffered saline with Tween 20 (TBST) for 60 min. The immunoblots were analysed using primary antibodies incubated overnight at 4 °C and secondary antibodies linked to horseradish peroxidase (HRP) (Cytiva) incubated for 1 hr at room temperature. After each step filters were washed four times with TBST. HRP signals were visualized by enhanced chemiluminescence (ECL) (Cytiva) (1:25000) and imaged with Chemidoc (Bio-Rad).

Primary antibodies targeting GAPDH (Chemicon, MONOCLONAL ANTIBODY374), Cldn5 (ThermoFischer Scientific, 352500), ZO-1 (ThermoFischer Scientific, 402200), Occludin (ThermoFischer, 711500), VE-Cadherin (R&D Systems, AF1002), JAM-A (*Martìn-Padura et al., 1998*), Cingulin (*Cardellini et al., 1996*) and ESAM (R&D Systems, AF2827) were used at a dilution of 1:1,000.

## Quantitative PCR

Lungs from control and *Cldn5* iECKO mice were removed into RNAlater (ThermoFischer Scientific, AM7024). RNA was extracted and purified using RNeasy Plus kit (Qiagen). RNA concentrations were measured by Nanodrop spectrophotometer (ThermoFisher Scientific) and adjusted to equal concentration, followed by reverse transcription using iScript Adv cDNA Kit for RT-qPCR (Bio-Rad, 1725038). Real-time quantitative PCRs were performed on Bio-Rad real-time PCR machine using the Taqman assay with the following probes: *Gapdh* (Mm99999915_g1), *Cldn5* (Mm00727012_s1), *Cdh5* (Mm00486938_m1), *Tjp1* (Mm01320638_m1), *Ocln* (Mm00500912_m1), *F11r* (Mm00554113_m1), *Cgn* (Mm01263534_m1), *Esam* (Mm00518378_m1). The comparative Ct method was used to calculate fold differences.

## Electron microscopy

To study junction ultrastructure, control or *Cldn5* iECKO mice were anesthetized and perfused first with 10 mL HBSS and then 12 mL cold fixative (1% GA, 4% PFA in 0.1 M phosphate buffer) through the left ventricle. To study HRP penetrance, control or *Cldn5* iECKO mice were intravenously injected with HRP (Sigma Aldrich, 77332) (800 mg/kg) and histamine (10 mg/kg) before anaesthetization and perfusion after 10 min. Ears were then removed and placed in fixative for 30 min at 4 °C before washing in PBS. Samples with HRP were treated with 0.05% DAB in PBS for 20 min and then with buffer containing $H_2O_2$ and 0.05% DAB for 30 min. Samples were post-fixed with 1% $OsO_4$ +1,5% $C_6FeK_4N_6$ 2 h at 4 °C before dehydration and LX112 resin infiltration. Post-staining was with $C_{12}H_{10}O_{14}Pb_3$. Samples without HRP were treated as above but devoid of DAB treatment and including 1% $C_4H_6O_6U$ (Uranyl acetate) staining. Samples with HRP were treated with 0.05% DAB in PBS for 20 min and then with buffer containing $H_2O_2$ and 0.05% DAB for 30 min. Samples were post-fixed with 1% $OsO_4$ +1,5% $C_6FeK_4N_6$ for 2 hr at 4 °C before dehydration and LX112 resin infiltration. Post-staining was with $C_{12}H_{10}O_{14}Pb_3$. Samples without HRP were treated as above but devoid of DAB treatment and including 1% $C_4H_6O_6U$ (Uranyl acetate) staining. Transmission electron microscopy (TEM) imaging and analysis was done at the Euro-BioImaging facility at Biocenter Oulu, Finland.

## Image quantification

For TEM analysis junction width was measured in the region bordered by electron dense junctional actin where the junction width is smallest. HRP penetrance was measured from the beginning of the junctional cleft on the luminal surface to the furthest point of HRP penetrance within the junctional space. Each value consisted of 1–3 measurements.

For confocal images macromolecular leakage was quantified by measurement of tracer area following image thresholding. Tracer area was then normalized to vessel area. Protein levels were quantified by measurement of their area within vessels following image thresholding. Areas were then normalized to vessel area. Analysis within GFP-positive or -negative vessels was conducted following the generation of a mask within the GFP channel. Similarly, analysis of αSMA-negative arterioles, capillaries and venules was done following generation of a mask in the αSMA channel. Arterioles could be separated from venules by their distinctive striated αSMA pattern. Threshold values were kept constant within experiments.

Analysis of leakage from intravital movies has been described previously (*Richards et al., 2021*). Briefly, leakage sites were identified in time-lapse imaging as defined sites of concentrated dextran in the extravascular space. To quantify these their numbers were normalized to vessel length. To assess lag period the time of the appearance of these sites following injection of stimulus was quantified. To assess the extent of barrier disruption the rate of dextran extravasation specifically at these sites was quantified over time.

All measurements were done with Fiji processing package of Image J2 software.

## Statistical analysis

Data are expressed as mean ± SD. The principal statistical test used was the Students' t test or One-way ANOVA with Tukey post-hoc test (multiple comparisons). p-*value*s given are from independent samples analysed by two-tailed paired t tests. Rate of leakage was compared using linear regression and ANCOVA. Correlations were calculated using a Pearson's correlation. All statistical analyses were conducted using GraphPad Prism. A p-value <0.05 was considered statistically significant and

significances indicated as p<0.05 (*), p<0.01 (**), and p<0.001 (***). For animal experiments no statistical methods were used to predetermine sample size. The investigators were blinded to allocation during experiment and outcome assessment.

## Acknowledgements

The authors acknowledge the Biocenter Oulu Electron Microscopy Core Facility supported by Biocenter Finland and the University of Oulu for their specific scientific expertise and research infrastructure services and the equipment and expert advice supplied by the BioVis imaging and flow cytometry core facility (Uppsala University). This study was supported by the Swedish Research Council (2020–01349), the Knut and Alice Wallenberg foundation (KAW 2020.0057 and KAW 2019.0276), Fondation Leducq Transatlantic Network of Excellence Grant in Neurovascular Disease (17 CVD 03) and the Swedish Cancer Society (Cancerfonden) 19 0119 Pj and 19 0118 Us to LC-W. KK and MG were supported by Cancerfonden (20 1,086 Pj), KK was supported by Wallenberg Academy Fellowship (2017.0144), Ragnar Söderbergs Fellowship (M13/17). ES was supported by Svenska Sällskapet för Medicinsk Forskning (SSMF). EN was supported by the Gustaf Adolf Johansson's foundation. SN was supported by Åke Wibergs foundation (M21-0109). MR was supported by SSMF (201912) and an EMBO long-term fellowship (ALTF 923–2016).

## Additional information

### Funding

| Funder | Grant reference number | Author |
| --- | --- | --- |
| Vetenskapsrådet | 2020-01349 | Lena Claesson-Welsh |
| Knut och Alice Wallenbergs Stiftelse | 2020.0057 | Lena Claesson-Welsh |
| Knut och Alice Wallenbergs Stiftelse | 2019.0276 | Lena Claesson-Welsh |
| Fondation Leducq | 17 CVD 03 | Lena Claesson-Welsh |
| Cancerfonden | 19 0119 Pj | Lena Claesson-Welsh |
| Cancerfonden | 19 0118 Us | Lena Claesson-Welsh |
| Cancerfonden | 20 1086 Pj | Katarzyna Koltowska Marleen Gloger |
| Knut och Alice Wallenbergs Stiftelse | 2017.0144 | Katarzyna Koltowska |
| Ragnar Söderbergs stiftelse | M13/17 | Katarzyna Koltowska |
| Åke Wiberg Stiftelse | M21-0109 | Sofia Nordling |
| Svenska Sällskapet för Medicinsk Forskning | | Elin Sjöberg |
| Svenska Sällskapet för Medicinsk Forskning | 201912 | Mark Richards |
| European Molecular Biology Organization | ALTF 923-2016 | Mark Richards |

The funders had no role in study design, data collection and interpretation, or the decision to submit the work for publication.

### Author contributions

Mark Richards, Conceptualization, Data curation, Formal analysis, Funding acquisition, Validation, Investigation, Visualization, Methodology, Writing – original draft, Project administration, Writing – review and editing; Emmanuel Nwadozi, Investigation, Methodology, Writing – review and editing;

Sagnik Pal, Pernilla Martinsson, Mika Kaakinen, Elin Sjöberg, Investigation; Marleen Gloger, Investigation, Writing – review and editing; Katarzyna Koltowska, Resources, Writing – review and editing; Christer Betsholtz, Lauri Eklund, Resources; Sofia Nordling, Conceptualization, Data curation, Formal analysis, Investigation, Visualization, Methodology, Writing – original draft, Writing – review and editing; Lena Claesson-Welsh, Conceptualization, Resources, Funding acquisition, Methodology, Writing – original draft, Project administration, Writing – review and editing

Author ORCIDs
Mark Richards ⓘ http://orcid.org/0000-0002-2266-3329
Sagnik Pal ⓘ http://orcid.org/0000-0002-5562-1555
Marleen Gloger ⓘ http://orcid.org/0000-0002-3319-7642
Elin Sjöberg ⓘ http://orcid.org/0000-0002-8799-4874
Katarzyna Koltowska ⓘ http://orcid.org/0000-0002-6841-8900
Lauri Eklund ⓘ http://orcid.org/0000-0002-3177-7504
Lena Claesson-Welsh ⓘ http://orcid.org/0000-0003-4275-2000

### Ethics

In vivo animal experiments were carried out in accordance with the ethical permit provided by the Committee on the Ethics of Animal Experiments of the University of Uppsala (permit 6789/18).

### Decision letter and Author response

Decision letter https://doi.org/10.7554/eLife.78517.sa1
Author response https://doi.org/10.7554/eLife.78517.sa2

## Additional files

### Supplementary files
• MDAR checklist

### Data availability

The murine ear skin data has been deposited in GEO under accession number GSE202290. Further details regarding specifics of the analysis will be available upon reasonable request.

The following dataset was generated:

| Author(s) | Year | Dataset title | Dataset URL | Database and Identifier |
|---|---|---|---|---|
| Richards M | 2022 | Claudin5 protects the peripheral endothelial barrier in an organ and vessel type-specific manner | http://www.ncbi.nlm.nih.gov/geo/query/acc.cgi?acc=GSE202290 | NCBI Gene Expression Omnibus, GSE202290 |

The following previously published datasets were used:

| Author(s) | Year | Dataset title | Dataset URL | Database and Identifier |
|---|---|---|---|---|
| Kalucka J | 2020 | Single-Cell Transcriptome Atlas of Murine Endothelial Cells | https://www.ebi.ac.uk/arrayexpress/experiments/E-MTAB-8077/ | ArrayExpress, E-MTAB-8077 |
| Pisco A | 2020 | A single-cell transcriptomic atlas characterizes ageing tissues in the mouse | https://doi.org/10.6084/m9.figshare.8273102.v2 | figshare, 10.6084/m9.figshare.8273102.v2 |
| The Tabula Muris Consortium | 2020 | A single-cell transcriptomic atlas characterizes ageing tissues in the mouse | https://www.ncbi.nlm.nih.gov/geo/query/acc.cgi?acc=GSE132042 | NCBI Gene Expression Omnibus, GSE132042 |

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
