## [Editor Report]

Understanding molecular and cellular mechanisms for endothelial cell (EC) barrier integrity at homeostasis and its impairment at pathologic conditions are fundamental and central topics for understanding vascular biology and related cardiovascular diseases. In this study, the authors elegantly demonstrated that the claudin5 deficiency enhances histamine‐induced leakage in organ- and vessel type‐specific, and size‐selective manners, which could be the result of alternative compositions of adherens and tight junctional proteins in the ECs. This study will aid our ability to modify EC barrier stability in a targeted, organ‐specific manner.

---

## [Decision Letter]

**Decision letter after peer review:**

Thank you for submitting your article "Claudin5 protects the peripheral endothelial barrier in an organ and vessel type-specific manner" for consideration by *eLife*. Your article has been reviewed by 3 peer reviewers, including Gou Young Koh as the Reviewing Editor and Reviewer #1, and the evaluation has been overseen Anna Akhmanova as the Senior Editor. The following individuals involved in the review of your submission have agreed to reveal their identity: Pipsa Saharinen (Reviewer #2).

Essential revisions:

The comments below recommend to the authors significant additional experimental works for this study. Please provide the reasons for not making some of the suggested changes where necessary.

1. The authors employed the histamine injection to see the differential vascular leakage at the different capillary beds. To ensure their claims, they are required to examine the vascular leakage in pathologic conditions such as injury or inflammation. Physical injury to the skin and ischemic injury to the skeletal muscle and myocardium could be feasible for the authors.

2. Gene expression is thoroughly investigated in the ear skin, back skin, trachea, skeletal muscle and heart. However, protein expression was not carefully examined. Claudin5 was still expressed even after the cldn5 gene was knocked out in endothelial cells. Therefore, immunohistochemistry of blood vessels should be carefully performed to confirm where the compensatory increase of gene expression is found.

3. Given the dominant expression of Claudin5 in arterioles over venules, leakage in the cldn5 KO mouse should be mostly found in arterioles. The authors did not show clear data about the leakage sites. Figure 4B vi and Figure 3C ii suggested the leakage in the capillaries and venules.

4. In Figure 3Fiii, how could it be explained that the extravasation of larger 200 nm microspheres is increased significantly in the back skin of Cldn5 iECKO mice when compared to controls, whereas the leakage of smaller molecular size dextran (2000 kda) is not? One would expect that smaller tracers would leak out similarly unless the leakage in the control is already saturating. This should be clarified as it concerns the authors' conclusion about the size-selective regulation of dermal vasculature by Cldn5.

5. Cldn5 iECKO lung lysates (For example Figure 3aii) show relatively high Cldn5 expression, which is explained by protein stability. The deletion time is 5 days post last tamoxifen. Have the authors tested longer deletions to get rid of the residual Cldn5 protein expression and does this affect the observed phenotype of e.g. the non-leaky arterioles in the Cldn5 iECKO mice? It is also mentioned that topical administration of tamoxifen was used "to enhance the loss of Claudin5 without causing lethality". It will be interesting to report if the deletion of Cldn5 in adult mice was lethal.

*Reviewer #1 (Recommendations for the authors):*

The questions and suggestions below are not intended to trigger significant additional experimental work for the present paper, but to perhaps increase clarity and make the study more accessible to readers.

1. The authors employed the histamine injection to see the differential vascular leakage at the different capillary beds. To ensure their claims, they are required to examine the vascular leakage in pathologic conditions such as injury or inflammation. Physical injury to the skin and ischemic injury to the skeletal muscle and myocardium could be feasible for the authors.

2. Although the authors claimed they studied the different peripheral organs, they examined three tissues – skin, muscle, and outer mucosae (trachea). In this regard, the title and some descriptions in the text should be justified with modifications.

3. The finding of substantially expressed claudin 13 (T-cadherin), claudin-1, claudin-15, ceacam1, amotl1, and amotl2 is interesting. More elaboration about the finding is required in the text.

4. The lack of apparent change in junction structure at ultrastructural levels, but the change in junction integrity, is interesting. As they have mentioned, it could be reasoned by compensatory changes in compositions of EC junctional proteins. Please discuss how it can occur – possible mechanisms

5. In Supple Figures 6A and 6D, please specify the portion of the tibialis anterior muscle and heart (left ventricle?). Convincing and highly magnified images with statistical analysis are required to support the authors' claim.

*Reviewer #2 (Recommendations for the authors):*

1. Figure 3Fiii. How could it be explained that the extravasation of larger 200 nm microspheres is increased significantly in the back skin of Cldn5 iECKO mice when compared to controls, whereas the leakage of smaller molecular size dextran (2000 kda) is not? One would expect that smaller tracers would leak out similarly unless the leakage in the control is already saturating. This should be clarified as it concerns the authors' conclusion about the size-selective regulation of dermal vasculature by Cldn5.

2. Why did the authors not choose to study the CNS vasculature, which has continuous capillary junctions and shows increased leakiness in the constitutive Cldn5 knockout? This refers to the authors' conclusion that "Claudin5... has a limited role in maintaining baseline EC barrier integrity in blood vessels outside of the CNS but is involved in the protection of vessels against agonist‐induced macromolecular leakage …" Since CNS was not investigated using smaller tracers, is it possible that the mature CNS vasculature would be less sensitive to Cldn5 loss than the neonatal vasculature?

3. Cldn5 iECKO lung lysates (For example Figure 3aii) show relatively high Cldn5 expression, which is explained by protein stability. The deletion time is 5 days post last tamoxifen. Have the authors tested longer deletions to get rid of the residual Cldn5 protein expression and does this affect the observed phenotype of e.g. the non-leaky arterioles in the Cldn5 iECKO mice? It is also mentioned that topical administration of tamoxifen was used "to enhance the loss of Claudin5 without causing lethality". It will be interesting to report if the deletion of Cldn5 in adult mice was lethal.

4. The following text in the results should be clarified: "We conclude that loss of Claudin5, to a large degree, enhances histamine‐induced leakage in vessels which are Claudin5 negative, according to immunohistochemistry as well as to the GFP reporter mouse used here."

5. Related to #4, the authors find using RNAscope Cldn5 expression in capillaries and postcapillary venules and conclude that "Claudin5 is responsible for limiting the disruption of EC junctions in particular in capillaries but also in immediately following venules." Please clarify, as gene expression data was used to conclude that loss or decreased Cldn5 expression towards capillary/venous side is responsible for permeability in these vessel types.

*Reviewer #3 (Recommendations for the authors):*

1) If the authors consider that the agonist (histamine)-induced leakage is dependent on Claudin5 expression in organs and vessel subsets, at least histamine receptors are evenly expressed in endothelial cells on organ vessels they tested. If the expression of receptors varies, the leakage must follow the expression pattern of receptors that determines the intracellular signaling for cell-cell adhesions. Thus, the expression of histamine receptors (H1 and H2) should be examined in the organs (ear skin, trachea, skeletal muscle and heart) to conclude that leakage is dependent on the expression of adhesion molecule instead of the strength of cytokine-dependent signaling.

2) Immunohistochemistry of Claudin5 should be performed in the ear skin where cldn5 gene expression diminishes along the arteriovenous axis. Claudin5 expression is stable as shown in Figure 3Aii even after its gene deletion. Thus, gene expression analyses, as well as immunohistochemistry, help define the localization of Claudin5 that regulates the vascular permeability in vivo.

---

## [Author Response]

Essential revisions:The comments below recommend to the authors significant additional experimental works for this study. Please provide the reasons for not making some of the suggested changes where necessary.1. The authors employed the histamine injection to see the differential vascular leakage at the different capillary beds. To ensure their claims, they are required to examine the vascular leakage in pathologic conditions such as injury or inflammation. Physical injury to the skin and ischemic injury to the skeletal muscle and myocardium could be feasible for the authors.

We appreciate the reviewers’ suggestion to address leakage parameters in pathological models of injury and disease; we agree this is an important next step. However, our aim in the current study was to identify vessel segment- and organ-specific leakage response to a specific agonist. This type of ambitious in vivo investigation has not been performed before and it reveals a considerable variability in the leakage parameters between vessel subtypes and between organs. We thus respectfully argue that adding models for example of ischemic injury of the skeletal muscle and heart with endogenous production of a potentially wide range of cytokines, would introduce quite complex incomparable patterns, rendering such data sets out of scope for this study. From attempting to introduce these models (which are not as yet established with us) we also realized that it will require quite some practise to induce the insults in a reproducible manner. We do, however, present new data on ozazolone-induced dermatitis comparing the responses in ear and back skin (Figure 5K and L), which shows a pattern of leakage similar to that established upon exogenous administration of histamine. Thus, following Oxazolone-induced dermatitis, there is a significant increase in 2000kDa dextran leakage in the ear skin of Cldn5 iECKO mice but not the back skin, in keeping with the lesser role of Cldn5 in the back skin, in response to histamine. Overall, these data support the relevance of the design of our study and motivates a new section in the discussion on leakage in pathologies affecting the organs in focus (lines 369-376). We thank the reviewer for this suggestion.

2. Gene expression is thoroughly investigated in the ear skin, back skin, trachea, skeletal muscle and heart. However, protein expression was not carefully examined. Claudin5 was still expressed even after the cldn5 gene was knocked out in endothelial cells. Therefore, immunohistochemistry of blood vessels should be carefully performed to confirm where the compensatory increase of gene expression is found.

We appreciate the need to confirm and examine protein expression and not rely only on gene expression. Therefore, we addressed protein expression to understand which vessel subtypes are affected in Figure 5. Given the lethality of removing Cldn5 from the vasculature (See essential revision 5) and the apparent stability of Cldn5 protein (Figure 3B-E and Figure 4—figure supplement 1 B and F), full removal of Cldn5 protein from all endothelial cells seems not possible with the available methodology. In Figure 3 and some of the revision experiments that we have now conducted, we show that there is still some remaining Cldn5 protein expression in arterioles, whereas Cldn5 is no longer visible in capillaries. In venules Cldn5 protein is not visible in the wildtype control situation, but we can detect some mRNA expression by RNAscope. Thus, for Cldn5, we do not claim that there is compensatory gene expression from some other cell type but that loss of the protein from endothelial cells is incomplete.

We do also address compensatory expression mechanisms at the gene and protein level for other important EC junction proteins in Figures 5 and Figure 5—figure supplement 1. Here we find that in response to the loss of Cldn5 there is upregulation of Occludin in the lung. We also find upregulation of VE-cadherin and downregulation of ZO-1 in both the lung and ear dermis, with these changes being present in capillaries and venules, whereas their expression in arterioles appears unaffected. To strengthen the data provided in our initial submission we have conducted further experiments on VE-cadherin and ZO-1 expression in vessel subtypes of the ear dermis following Cldn5 knock-out and added analysis in the back skin, where we see no effect of Cldn5 loss of the expression of VE-cadherin and ZO-1.

3. Given the dominant expression of Claudin5 in arterioles over venules, leakage in the cldn5 KO mouse should be mostly found in arterioles. The authors did not show clear data about the leakage sites. Figure 4B vi and Figure 3C ii suggested the leakage in the capillaries and venules.

In Figure 4A and 4B we aimed to demonstrate that we do not observe leakage occurring from arterioles in neither the wildtype control nor the Cldn5 iECKO. We have now added additional analysis in Figure 4H, showing comparison of extravascular dextran observed in association with aSMA-positive arterioles, aSMA-negative capillaries and aSMA-positive venules. This shows the minimal level of dextran signal that is associated with arterioles compared to capillaries and venules, regardless of Cldn5 levels.

However, we cannot state definitively that Cldn5-deficient arterioles cannot respond to histamine with increased leakage. As we point out, we cannot fully remove Cldn5 protein from arterioles: even when tamoxifen was administered topically on the ear skin (to avoid sudden death of the animal) and waiting up to 30 days for remaining protein to be cleared, there was still detectable levels of Cldn5 in arterioles (see Figure 4—figure supplement 1). Small levels of residual Cldn5 may thus allow ECs to maintain sufficient junctional integrity to prevent the leakage of 2000kDa Dextran. Cldn5 in arterioles may also be compensated by other proteins such as Claudin15, which we also see is more highly expressed in arterioles. We have, furthermore, now included data describing histamine receptor expression (Figure 4—figure supplement 1D) to exclude that the lack of leakage from arterioles would be due to reduced receptor expression. The histamine receptors, Hrh1 and Hrh2, are uniformly expressed among the ear dermal subsets or absent respectively in our scRNAseq analysis indicating that the arterio-venous differences in leakage is not due to the lack of receptor expression in the arterioles.

4. In Figure 3Fiii, how could it be explained that the extravasation of larger 200 nm microspheres is increased significantly in the back skin of Cldn5 iECKO mice when compared to controls, whereas the leakage of smaller molecular size dextran (2000 kda) is not? One would expect that smaller tracers would leak out similarly unless the leakage in the control is already saturating. This should be clarified as it concerns the authors' conclusion about the size-selective regulation of dermal vasculature by Cldn5.

An important difference between the 2000 kDa dextran tracer and the 200 nm microspheres is in their dimensions, with the dextran being an extended, rather flexible structure while the microsphere is a static globular entity. In our new analyses of the effects of lower histamine concentrations the difference between the leakage of dextran and microspheres in the Cldn5-deficient condition, was not maintained. Instead, we see an inversion of the phenotype, with the back skin and trachea now having increased leakage to 2000 kDa dextran and not microspheres. We believe that the dynamics and dimensions of the leakage sites can explain the difference in leakage properties of the two tracers, however, this is currently not possible to prove. We have therefore decided to remove the microsphere data from the edited version but are providing for the reviewers graphs of 2000 kDa dextran and microsphere leakage in trachea and back skin for high (10 mg/kg) and low (4 mg/kg) histamine doses for you to observe the different phenotypes (Please see reviewer figure 1).

5. Cldn5 iECKO lung lysates (For example Figure 3aii) show relatively high Cldn5 expression, which is explained by protein stability. The deletion time is 5 days post last tamoxifen. Have the authors tested longer deletions to get rid of the residual Cldn5 protein expression and does this affect the observed phenotype of e.g. the non-leaky arterioles in the Cldn5 iECKO mice? It is also mentioned that topical administration of tamoxifen was used "to enhance the loss of Claudin5 without causing lethality". It will be interesting to report if the deletion of Cldn5 in adult mice was lethal.

The deletion of Cldn5 in adult mice is indeed lethal. This data is being prepared for publication by Christer Betsholtz’s lab. Mean humane endpoints in this model are typically around 12 days post induction. For this reason, we have conducted experiments 5 days post-tamoxifen to avoid any problems related to mouse health. We have used topical tamoxifen as an alternative to attempt to push the deletion further in the ear dermis, however as we report in Figure 3 and Figure 4—figure supplement 1 we still see a similar phenotype using systemic or topical tamoxifen treatment.

To try and push topical tamoxifen treatment further we are also now including 30 day post topical tamoxifen data with 2 x 3 doses of hydroxy-tamoxifen instead of the previous 1 x 3 doses. We have also investigated histamine-induced leakage in these mice (See Figure 4—figure supplement 1). From these experiments we find that we are able to induce a higher loss of Cldn5 protein (remaining levels, 13% of control). Analysis of histamine-induced leakage in these mice however shows similar response to those systemically treated with tamoxifen. Thus, we still do not observe any obvious arteriolar leakage in topically treated Cldn5 iECKO mice. We argue that the remaining 13% expression level of Cldn5 in arterioles is sufficient to maintain the barrier to 2000 kDa dextran or that other aspects of the EC junctional complex provide sufficient barrier integrity in the absence of Cldn5.

Reviewer #1 (Recommendations for the authors):The questions and suggestions below are not intended to trigger significant additional experimental work for the present paper, but to perhaps increase clarity and make the study more accessible to readers.1. The authors employed the histamine injection to see the differential vascular leakage at the different capillary beds. To ensure their claims, they are required to examine the vascular leakage in pathologic conditions such as injury or inflammation. Physical injury to the skin and ischemic injury to the skeletal muscle and myocardium could be feasible for the authors.

Please see above under essential revisions point 1.

2. Although the authors claimed they studied the different peripheral organs, they examined three tissues – skin, muscle, and outer mucosae (trachea). In this regard, the title and some descriptions in the text should be justified with modifications.

We would respectfully disagree with this comment. We do not claim that we have studied all tissues, but that the selected peripheral tissues with continuous endothelium selected for our analyses show differences in their endothelial barrier properties and that Cldn5 has different protective roles in them. This we believe we show given the different phenotypes in the studied tissues and thus we feel that we are justified in claiming that Cldn5 offers organotypic and vessel specific barrier protection mechanisms.

3. The finding of substantially expressed claudin 13 (T-cadherin), claudin-1, claudin-15, ceacam1, amotl1, and amotl2 is interesting. More elaboration about the finding is required in the text.

More attention has now been given to other genes in the text. Please see lines 124-145.

4. The lack of apparent change in junction structure at ultrastructural levels, but the change in junction integrity, is interesting. As they have mentioned, it could be reasoned by compensatory changes in compositions of EC junctional proteins. Please discuss how it can occur – possible mechanisms

We have now added more detailed discussion around these points. Please see lines 390-403.

5. In Supple Figures 6A and 6D, please specify the portion of the tibialis anterior muscle and heart (left ventricle?). Convincing and highly magnified images with statistical analysis are required to support the authors' claim.

We have added information in the Methods section for the TA muscle and heart. Please see lines 750-757. Regarding higher magnified images, we feel that the magnified inserts in Supp Figure 6A (now Figure 2—figure supplement 1) are sufficient to prove our claim that extravasated dextran signal is present in the capillaries of skeletal muscle and that the heart vasculature is refractory to histamine-induced leakage. We have added magnified inserts for Supp Figure 6D (Now Figure 4—figure supplement 1E). We have also supplied non-normalised values for 2000kDa dextran leakage in all organs stimulated with a high dose of histamine in Figure 3—figure supplement 1B as further proof that the heart vasculature is resistant histamine-induced leakage. Here the heart shows at least a 100-fold reduction in fluorescent signal compared to other tissue.

Reviewer #2 (Recommendations for the authors):1. Figure 3Fiii. How could it be explained that the extravasation of larger 200 nm microspheres is increased significantly in the back skin of Cldn5 iECKO mice when compared to controls, whereas the leakage of smaller molecular size dextran (2000 kda) is not? One would expect that smaller tracers would leak out similarly unless the leakage in the control is already saturating. This should be clarified as it concerns the authors' conclusion about the size-selective regulation of dermal vasculature by Cldn5.

Please see essential revisions point 4 above.

2. Why did the authors not choose to study the CNS vasculature, which has continuous capillary junctions and shows increased leakiness in the constitutive Cldn5 knockout? This refers to the authors' conclusion that "Claudin5 … has a limited role in maintaining baseline EC barrier integrity in blood vessels outside of the CNS but is involved in the protection of vessels against agonist‐induced macromolecular leakage …" Since CNS was not investigated using smaller tracers, is it possible that the mature CNS vasculature would be less sensitive to Cldn5 loss than the neonatal vasculature?

As mentioned above, work on the CNS is being conducted in the lab of Christer Betsholtz and the CNS Cldn5 iECKO mouse phenotype will shortly be published.

3. Cldn5 iECKO lung lysates (For example Figure 3aii) show relatively high Cldn5 expression, which is explained by protein stability. The deletion time is 5 days post last tamoxifen. Have the authors tested longer deletions to get rid of the residual Cldn5 protein expression and does this affect the observed phenotype of e.g. the non-leaky arterioles in the Cldn5 iECKO mice? It is also mentioned that topical administration of tamoxifen was used "to enhance the loss of Claudin5 without causing lethality". It will be interesting to report if the deletion of Cldn5 in adult mice was lethal.

Please see our response in essential revisions point 5.

4. The following text in the results should be clarified: "We conclude that loss of Claudin5, to a large degree, enhances histamine‐induced leakage in vessels which are Claudin5 negative, according to immunohistochemistry as well as to the GFP reporter mouse used here."

This part of the text has been modified. See lines 294-301. We hope that this is more appropriate.

5. Related to #4, the authors find using RNAscope Cldn5 expression in capillaries and postcapillary venules and conclude that "Claudin5 is responsible for limiting the disruption of EC junctions in particular in capillaries but also in immediately following venules." Please clarify, as gene expression data was used to conclude that loss or decreased Cldn5 expression towards capillary/venous side is responsible for permeability in these vessel types.

Here we were referring to Claudin5 in wild-type vessels. We have clarified this. See line 303

Reviewer #3 (Recommendations for the authors):1) If the authors consider that the agonist (histamine)-induced leakage is dependent on Claudin5 expression in organs and vessel subsets, at least histamine receptors are evenly expressed in endothelial cells on organ vessels they tested. If the expression of receptors varies, the leakage must follow the expression pattern of receptors that determines the intracellular signaling for cell-cell adhesions. Thus, the expression of histamine receptors (H1 and H2) should be examined in the organs (ear skin, trachea, skeletal muscle and heart) to conclude that leakage is dependent on the expression of adhesion molecule instead of the strength of cytokine-dependent signaling.

We thank the reviewer for their suggestion to examine histamine receptor expression. We have now included data describing histamine receptor expression (text line 260; Figure 4—figure supplement 1D). The histamine receptor largely responsible for EC barrier disruption (Hrh1) is not significantly differentially expressed among the subsets in our scRNAseq analysis and appears evenly distributed in the ear vasculature. The other histamine receptor Hrh2 meanwhile, which has also been reported alongside vascular permeability only shows expression in some subsets of skeletal muscle and heart. Thus, we feel that this indicates that the arterio-venous differences in leakage is not due to insufficient receptor expression in the arterioles.

2) Immunohistochemistry of Claudin5 should be performed in the ear skin where cldn5 gene expression diminishes along the arteriovenous axis. Claudin5 expression is stable as shown in Figure 3Aii even after its gene deletion. Thus, gene expression analyses, as well as immunohistochemistry, help define the localization of Claudin5 that regulates the vascular permeability in vivo.

We now include an analysis of Cldn5 protein levels in different vessel subtypes in ear skin of control and Cldn5 iECKO mice (Figure 4—figure supplement 1F). We see that some residual level of Cldn5 immunostaining remains in aSMA-positive arterioles whilst Cldn5 is essentially gone in aSMA-negative capillaries. In aSMA-positive venules Cldn5 protein levels are non-existent in both control and Cldn5 iECKO mice.